# Extracellular ATP elicits DORN1-mediated RBOHD phosphorylation to regulate stomatal aperture

Dongqin Chen [1], Yangrong Cao [1,2], Hong Li[2], Daewon Kim[1], Nagib Ahsan[3,4], Jay Thelen[3] & Gary Stacey[1,3]

In addition to acting as a cellular energy source, ATP can also act as a damage-associated molecular pattern in both animals and plants. Stomata are leaf pores that control gas exchange and, therefore, impact critical functions such as photosynthesis, drought tolerance, and also are the preferred entry point for pathogens. Here we show the addition of ATP leads to the rapid closure of leaf stomata and enhanced resistance to the bacterial pathogen *Psuedomonas syringae*. This response is mediated by ATP recognition by the receptor DORN1, followed by direct phosphorylation of the NADPH oxidase RBOHD, resulting in elevated production of reactive oxygen species and stomatal closure. Mutation of DORN1 phosphorylation sites on RBOHD eliminates the ability of ATP to induce stomatal closure. The data implicate purinergic signaling via DORN1 in the control of stomatal aperture with important implications for the control of plant photosynthesis, water homeostasis, pathogen resistance, and ultimately yield.

[1] Division of Plant Science, C.S. Bond Life Science Center, University of Missouri, Columbia, MO 65211, USA. [2] State Key Lab of Agricultural Microbiology, College of Life Science and Technology, Huazhong Agricultural University, Wuhan 430070, China. [3] Division of Biochemistry, C.S. Bond Life Science Center, University of Missouri, Columbia, MO 65211, USA. [4]Present address: Center for Cancer Research Development, Proteomics Core Facility, Rhode Island Hospital,, Providence, RI 02903, USA. Correspondence and requests for materials should be addressed to G.S. (email: staceyg@missouri.edu)

All known organisms use adenosine 5′-triphosphate (ATP) as an essential, cellular energy source to drive many biochemical reactions. However, ATP can also be released into the extracellular matrix, where it is referred to as extracellular ATP (eATP), either in response to environmental stimuli or wounding, where it is recognized as a damage-associated molecular pattern (DAMP)[1,2]. For example, the release of eATP in animals can evoke a variety of responses, such as inflammation, neurotransmission, muscle contraction, and even cell death[1,3,4]. In mammalian cells, two classes of plasma membrane localized purinoceptors are known: P2X ligand-gated ion channels and P2Y G protein-coupled receptors that bind eATP-inducing complex, downstream cellular signaling pathways[5].

In contrast to mammals, the role of ATP as an extracellular signal in plants is poorly understood. However, a number of recent reviews have highlighted a variety of plant cellular processes that can be impacted by the addition of exogenous ATP[2,6–8]. For example, eATP has been implicated in root hair growth[9,10], biotic and abiotic stress responses[11–13], gravitropism[14], cell death[15,16], and thigmotropism[17]. Plants respond to eATP with many of the same responses seen in animals, including elevation of cytoplasmic calcium [Ca$^{2+}$]$_{cyt}$[18], production of nitric oxide (NO)[19,20] and reactive oxygen species (ROS)[9,19], mitogen-activated protein kinase (MAPK) phosphorylation[7], and expression of defense-related genes[7]. These various responses to eATP lacked a mechanistic explanation until the identification of the first plant purinoreceptor, DORN1 (DOESN'T RESPOND TO NUCLEOTIDES)[7], was reported in 2014, defining a new family of kinase purinergic receptor (i.e., P2K1). DORN1 encodes LecRK-I.9 (L-type lectin receptor-like kinase) composed of an extracellular legume L-type lectin domain, a transmembrane domain, and an intracellular serine/threonine kinase domain. The DORN1 extracellular binding domain binds to ATP and ADP with high affinity and to other purine nucleotides with lesser affinity[2,8]. Ectopic overexpression of DORN1 increased the plant response to physical wounding[7].

Previous studies identified LecRK-I.9 (DORN1) as a positive regulator of plant defense against the oomycete pathogens, *Phytophthora brassicae*, *Phytophthora infestans*, and bacterial pathogen *Pseudomonas syringae* DC3000[21–25]. The main port of entry by pathogenic bacteria, such as *P. syringae*, is via the leaf stomata. Hence, closure of the stomata reduces bacterial infection, but also restricts water vapor and gas exchange between the leaf and atmosphere[26,27]. The rapid generation of ROS has been shown to be among the initial plant responses to pathogen infection. However, ROS also plays a variety of important roles in plant growth and development: such as supporting cell wall growth and also stomatal aperture[28,29].

In *Arabidopsis*, ROS production in response to pathogen recognition (e.g., due to recognition of pathogen-associated molecular patterns (PAMPs)) is mediated by the NADPH oxidase resiratory burst oxidase homolog D (AtRBOHD) with AtRBOHF showing some redundant function[30–32]. AtRBOHD has FAD- and NADPH-binding sites in the C-terminal domain, six transmembrane domains and Ca$^{2+}$-binding EF-hand motifs in the N-terminal domain[33]. Previous studies showed that AtRBOHD function is mainly regulated through Ca$^{2+}$$_{cyt}$ concentration via direct binding to the EF-hand motifs. Subsequent activation of calcium-dependent protein kinases (CPKs or CDPKs), especially CPK5, results in activation of AtRBOHD by direct phosphorylation[34,35]. However, recent studies have disclosed a Ca$^{2+}$-independent regulation of AtRBOHD through direct activation via phosphorylation by Botrytis-induced kinase1 (BIK1), which itself is activated by direct interaction with PAMP-responsive pattern recognition receptors (PRRs)[36,37]. CPK5 and BIK1 phosphorylate AtRBOHD at unique sites, consistent with their role in independently regulating AtRBOHD activity.

Clark et al.[6] previously reported that exogenous addition of ATP would promote stomatal closure in the light but stimulated stomatal opening in darkness. However, no specific mechanism was offered to explain how ATP affected stomatal aperture. In the current study, we provide a molecular mechanism for the light response by showing that ATP-triggered activation of the DORN1 kinase function leads to direct phosphorylation of AtRBOHD, resulting in an elevation of ROS production and subsequent stomatal closure. This activity of DORN1 is important for the ability of exogenous ATP to enhance resistance to *P. syringae* infection.

## Results

**DORN1 autophosphorylation is vital for downstream signaling**. Our previous publication showed that DORN1 possesses an active, intracellular kinase domain and this activity is essential for ATP-induced plant responses (e.g., gene expression or elevation of Ca$^{2+}$$_{cyt}$ concentration)[7]. In order to determine the specific DORN1 residues critical for phosphorylation activity, we used mass spectrometry to identify a total of 13 autophosphorylation sites (Fig. 1a, Supplementary Data 1, and Supplementary Table 1). Supplementary Fig. 1a shows one example of this analysis, where the MS/MS-spectrum of the phosphopeptide QFVAEVVS(p)MR provides clear evidence for phosphorylation of DORN1 residue Ser391. In order to determine which of these phosphorylation sites are critical for DORN1 function, we mutated each site to either alanine (phosphorylation negative) or aspartate (phosphomimic). We expressed the mutant forms of DORN1 from the native promoter in the *dorn1-3* mutant background and examined each plant for the ability to respond to ATP addition by assaying changes in Ca$^{2+}$$_{cyt}$ concentration using aequorin luminescence[7]. As shown in Fig. 1b–d, the S391A, S440A, and S451A mutants when expressed in planta failed to complement the *dorn1-3* mutant phenotype. In contrast, the phosphomimic S391D, S440D, and S451D mutants exhibited a significant increase in Ca$^{2+}$$_{cyt}$ concentration in response to ATP addition (Fig. 1b–d and Supplementary Fig. 2a). These data suggest that autophosphorylation of S391, S440, and S451 is critical for mediating ATP signaling via the DORN1 kinase domain. Analysis of the plants expressing each of the other mutant proteins (either the A or D form) provided no evidence that these residues play a role in DORN1-mediated ATP signaling (Supplementary Fig. 2b, c).

PSM-based quantitative mass spectrometry showed that Ser391, Ser451, and Ser713 were highly phosphorylated. However, phosphorylation of Ser391 and Ser713 was predominant in the presence of Mg$^{2+}$ ion, whereas Ser440 and Ser715 were more highly phosphorylated in presence of Mn$^{2+}$ ion (Supplementary Fig. 1b). Meanwhile, all of these sites were phosphorylated when both Mg$^{2+}$ and Mn$^{2+}$ ions were present. Interestingly, addition of ATP changed the phosphorylation status for some sites (Supplementary Fig. 1b), consistent with ATP modulating DORN1 phosphorylation status.

As part of control experiments, we conducted immunoprecipitation experiments of the plants expressing each of the various mutant proteins to insure that each protein was expressed and at roughly equal intensity (Supplementary Fig. 2). Interestingly, when we co-expressed HA-tagged DORN1 and Myc-tagged DORN1 proteins in *Arabidopsis* and conducted co-immunoprecipitation (Co-IP) assays, DORN1 protein was shown to self-associate with the level of self-association enhanced upon ATP (Fig. 1e and Supplementary Fig. 8).

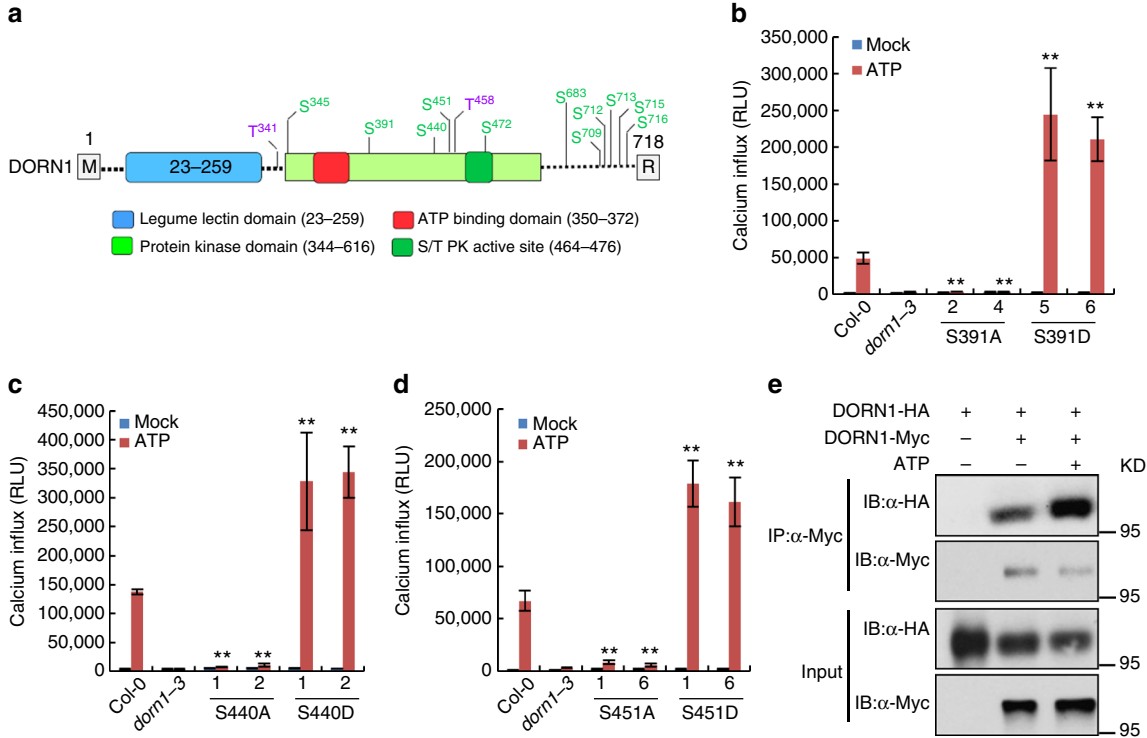

**Fig. 1** Mapping of DORN1 autophosphorylation sites and importance to DORN1 function. **a** Schematic representation of DORN1 protein structure highlighting the autophosphorylation sites identified by mass spectrometry. The various domains of DORN1 are color coded. **b**–**d** Contribution of different DORN1 autophosphorylation sites to ATP-induced calcium influx. Different transgenic plants (5-day-old) expressing the aequorin reporter were treated with 100 μM ATP, and the luminescence was immediately monitored. RLU relative luminescence units; Error bars indicate ± SEM; $n = 8$ (biological replicates); **$P < 0.01$, Student's $t$ test. These experiments were repeated three times with similar results. **e** Self-association of DORN1 in *Arabidopsis* protoplasts. DORN1-HA and DORN1-Myc were co-expressed in WT protoplasts treated with either 200 μM ATP for 20 min (+) or $H_2O$ as a control (−). Co-IP was performed using an anti-HA and anti-Myc antibodies. This experiment was repeated three times with similar results

**RBOHD is a kinase substrate of DORN1.** The data above suggest that DORN1 likely mediates purinergic signaling via ATP-induced phosphorylation of downstream target proteins. In order to identify such proteins, we used a mass spectrometry-based in vitro phosphorylation strategy, termed kinase client assay (KiC assay)[38]. A library of more than 2100 peptides developed from identified phosphorylation sites taken from a number of studies was incubated with recombinant GST-DORN1-KD kinase domain in the present of ATP. This mixture was then analyzed by mass spectrometry. Two sets of empty vectors (GST and MBP) and two inactive, kinase domain mutations GST-DORN1-1 (D572N) and GST-DORN1-2 (D525N) were used as negative controls. The results identified 23 phosphorylated peptides based on phosphoRS score and phosphoRS site probability. One of the most interesting of these proteins was RBOHD, where the synthetic peptide GILRGANS(p)DT(p)NSDTESI was phosphorylated by DORN1-KD in the KiC assay (Fig. 2a).

In order to extend this observation, we verified the DORN1–RBOHD interaction by Co-IP assays in wild-type protoplasts, where a HA-tagged DORN1 was co-expressed with a Myc-tagged RBOHD either with or without the addition of ATP (Fig. 2b and Supplementary Fig. 8). The data showed a clear interaction between DORN1 and RBOHD that was enhanced by ATP. As a negative control, the full-length chitin elicitor receptor kinase (CERK1)[39,40], a PRR essential for recognition of chitin as a PAMP did not interact with RBOHD.

The interaction of DORN1 with RBOHD suggested that ATP likely mediates ROS production through DORN1 kinase activity. Therefore, we tested the intensity of the ATP-induced ROS burst in 4–5-week-old plant leaves. Wild-type Col-0 plants showed a high ROS burst while, as expected, ROS generation was completely lost in *rbohd* mutant plants but was also significantly reduced in *dorn1-3* mutant plants (Fig. 2c and Supplementary Fig. 3a). Furthermore, while the expression of *RBOHD* and *DORN1* was induced 15 min after ATP treatment, *RBOHD* gene expression was significantly reduced in *dorn1-3* mutant plants (Supplementary Fig. 3b, c). These data are consistent with our previously published transcriptome data in which the *dorn1-3* mutant showed a marked reduction of RBOHD gene expression upon ATP treatment, but was induced more strongly in plants ectopically over-expressing the DORN1 protein[7]. However, in contrast to RBOHD mRNA, there was no concomitant increase in RBOHD protein from 0 to 30 min after ATP treatment (Supplementary Fig. 3d). These results are consistent with a role for DORN1 in direct activation of RBOHD resulting in an elevation of ROS production upon ATP addition.

**DORN1 directly interacts with RBOHD in planta.** The Co-IP results in Fig. 2b show the potential of DORN1 and RBOHD to form a complex, but do not provide unequivocal proof that these two proteins can interact directly in the absence of additional, unknown components. Therefore, we conducted in vitro pull-down experiments using GST-tagged, recombinant proteins purified from *E. coli* in which the N-terminal cytoplasmic domain of RBOHD (RBOHD-N) was able to bind to both wild-type and kinase inactive forms of the DORN1 kinase domain (DORN1-KD) (Fig. 3a and Supplementary Fig. 8). As further evidence that DORN1–RBOHD interaction is independent of DORN1 autophosphorylation, we treated the protein with lambda protein

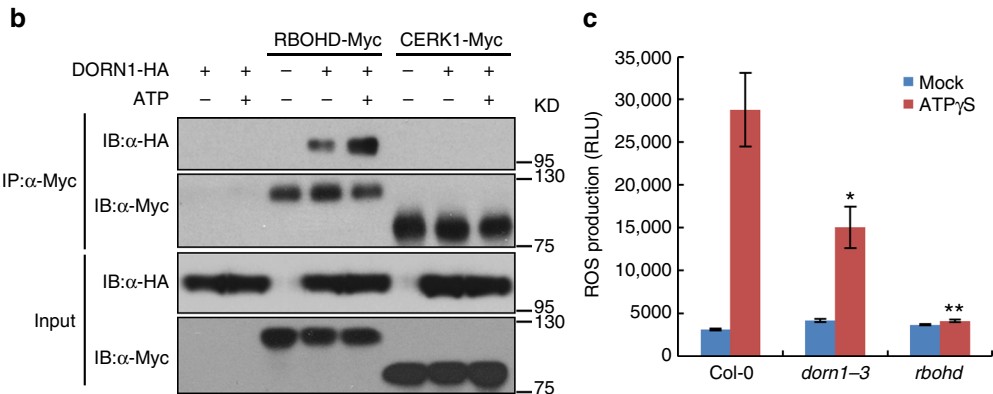

**Fig. 2** ATP triggers ROS response through DORN1 and RBOHD. **a** Identification of RBOHD tryptic peptides as a substrate of DORN1 kinase by KiC assay. GST-DORN1-KD kinase domain was incubated with a 2.1 K peptides library in the presence of ATP. The library was also incubated with the kinase-dead version of DORN1-KD-1, GST, or MBP as negative controls. Potential phosphorylation sites were predicted by phosphoRS. **b** Co-immunoprecipitation of DORN1 and RBOHD proteins in *Arabidopsis* protoplasts. The indicated constructs were transiently expressed in wild-type protoplasts treated with either 200 μM ATP for 20 min (+) or $H_2O$ as a control (−). Full-length CERK1 was used as a negative control. Co-IP was performed using an anti-HA and anti-Myc antibodies. This experiment was repeated three times with similar results. **c** DORN1 and RBOHD are important for the ATP-induced ROS burst. ROS production was measured using wild-type or *dorn1-3* and *rbohd* mutant plants treated with 250 μM ATPγS for 30 min. RLU relative luminescence units; values represent the mean ± SEM, $n = 8$ (biological replicates); *$P < 0.05$, **$P < 0.01$, Student's *t* test. This experiment was repeated three times with similar results

phosphatase to release any phosphate groups (Fig. 3a and Supplementary Fig. 8). Such treatment failed to block DORN1–RBOHD interaction. The intracellular kinase domain of the chitin-binding protein, LYK5[39], was used as a negative control in these assays and showed no binding to DORN1.

In order to further confirm the interaction of DORN1 with RBOHD, we conducted bimolecular fluorescence complementation (BiFC) assays in *Arabidopsis* protoplasts. The combination of DORN1-YFP[n] with RBOHD-YFP[c] produced a yellow fluorescent signal and merged with the signal of the plasma membrane marker FM4-64 (Fig. 3b), indicating that RBOHD specifically interacts with DORN1 on the plasma membrane. Furthermore, we also confirmed the results using in planta co-IP experiments. In this case, *DORN1-HA* was expressed under the control of the *DORN1* native promoter and *RBOHD-Myc* from the constitutive CaMV 35 s promoter in transgenic *Arabidopsis* plants (Supplementary Fig. 3e). ATP addition was found to significantly enhance DORN1–RBOHD interaction (Fig. 3c and Supplementary Fig. 8). Together, the results suggest that DORN1 directly interacts with RBOHD in a manner independent of DORN1 kinase activity.

**DORN1 phosphorylates RBOHD to promote ROS production.** The data above suggest that RBOHD is a direct phosphorylation target of the DORN1 kinase domain. In order to provide additional evidence in support of this hypothesis, we incubated N-terminal, recombinant RBOHD protein with the purified DORN1 kinase domain and assayed for phosphorylation by radiolabeling with $^{32}P$-ATP. Consistent with the KiC assay results, DORN1-KD strongly trans-phosphorylated RBOHD-N in vitro, whereas the two kinase-dead versions of DORN1-KD-1 and DORN1-KD-2 failed to phosphorylate RBOHD (Fig. 4a). In addition, we

generated RBOHD S22A, T24A, and S22AT24A recombinant proteins using PCR-based site-directed mutagenesis and incubated these proteins with purified DORN1-KD. The results showed a marked reduction in phosphorylation of these three mutant RBOHD proteins (Fig. 4b).

Importantly, the RBOHD S22 residue was previously reported to be phosphorylated based on in vivo phosphoproteomic analysis but no corresponding kinase was identified[41]. In order to verify the RBOHD phosphorylation sites resulting from DORN1 in vivo, we generated the RBOHD, S22A, T24A, and S22AT24A proteins fused with HA tag and transformed each into *rbohd* mutant protoplasts and subsequently analyzed their phosphorylation state. The wild-type RBOHD protein was phosphorylated in protoplasts upon ATP addition (Fig. 4c and Supplementary Fig. 8). In contrast, the T24 mutant protein showed a lower phosphorylation level than the wild-type protein while the S22A and S22AT24A mutant proteins showed very low levels of phosphorylation (Fig. 4c and Supplementary Fig. 8). The above data indicate that DORN1 directly phosphorylates RBOHD at S22 and T24 sites in vitro and in vivo.

In order to test the relevance of this phosphorylation to RBOHD-mediated ROS production, we measured the transient ROS burst by modification of a previously described procedure[42]. As shown in Fig. 4d, transient expression of RBOHD-HA complemented the loss of ATP-induced ROS production exhibited in *rbohd* mutant plants. However, the expression of the S22A mutant RBOHD protein significantly reduced the ability to restore ATP-induced ROS production in *rbohd* mutant plants, while expression of the S22D mutant resulted in enhanced ROS production (Fig. 4d and Supplementary Fig. 8). In contrast, there was no significant difference between the ability of either the T24A or T24D mutant proteins to complement the *rbohd* mutant (Fig. 4d and Supplementary Fig. 8), indicating that S22 is the

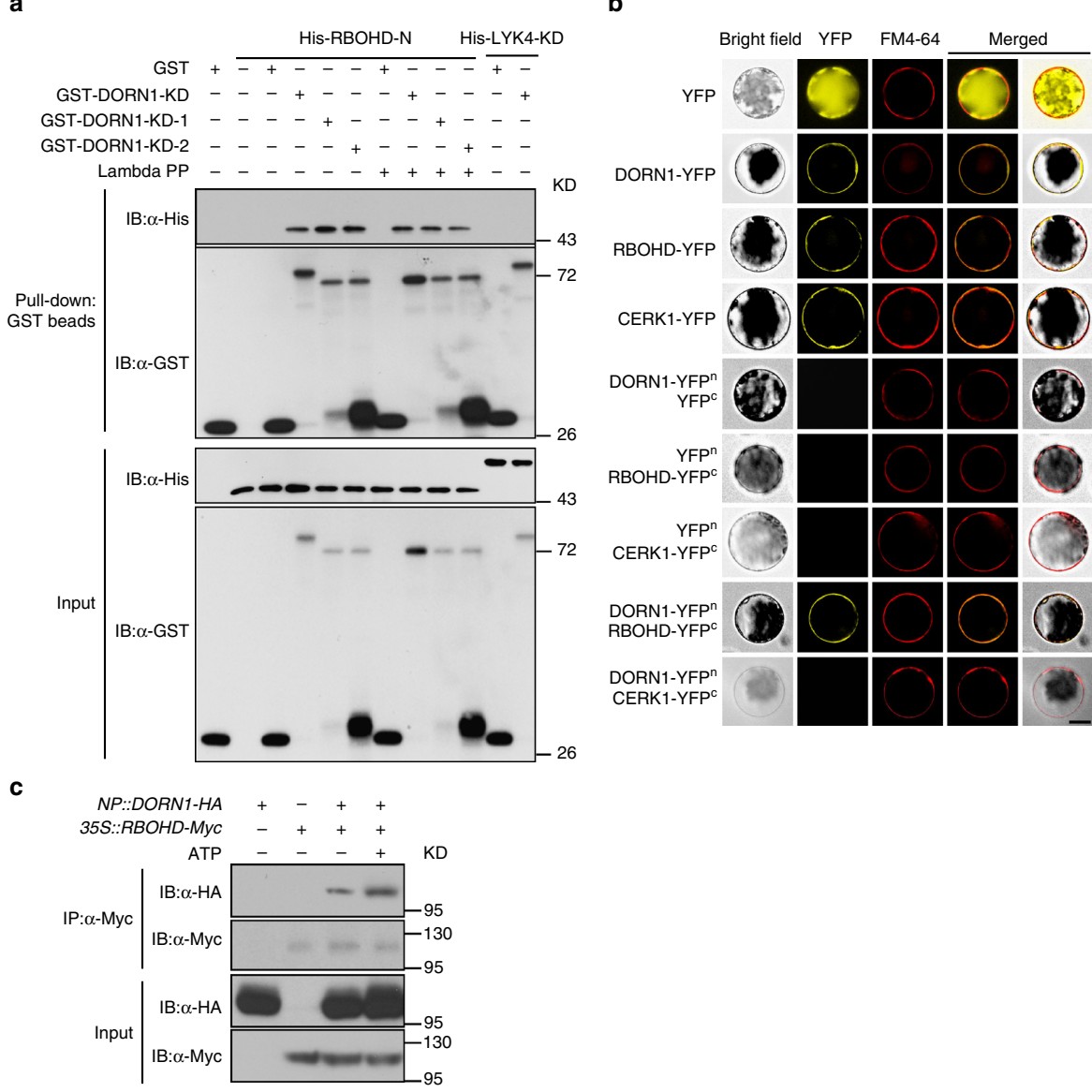

**Fig. 3** DORN1 directly interacts with RBOHD in vivo and in vitro. **a** DORN1 directly interacts with RBOHD N-terminal in vitro. Purified, recombinant proteins GST-DORN1-KD, GST-DORN1-KD-1 (kinase dead), GST-DORN1-KD-2 (kinase dead), or GST were incubated with His-RBOHD-N followed by GST-mediated pull-down. Purified His-Lyk4 kinase domain protein was used as a negative control. Lambda protein phosphatase (Lambda PP) was added to release phosphate groups from phosphorylated serine, threonine, and tyrosine residues. **b** DORN1 interacts with RBOHD in the protoplast plasma membrane. The indicated constructs were transiently expressed in wild-type protoplasts and the BiFC assay was performed. FM4-64 was added to stain the plant cell plasma membrane. Bar = 20 μm. **c** DORN1 interacts with RBOHD in *Arabidopsis* plants. Stable transgenic *Arabidopsis* plants (F1) generated from a cross between *NP::DORN1-HA/dorn1-3* and *35s::RBOHD-Myc/*Col-0 expressing lines were treated with or without 250 μM ATP for 30 min, and total protein extract was subjected to Co-IP. All above experiments were repeated three times with similar results

critical RBOHD residue for the ATP-induced ROS production. In order to confirm the specificity of the S22 site of RBOHD to DORN1 activation, we measured the flg22 and chitin-elicited ROS burst in S22A and S22D transgenic plants (Supplementary Figs. 4 and 5d). In the case of both treatments, the response of the wild type and the S22 phospho-null and phosphomimic were identical[41].

**DORN1 regulates stomatal closure and bacterial defense.** During pathogen invasion, activation of the RBOHD-dependent ROS burst induces stomatal closure effectively blocking entry of the pathogen to the leaf interior[36,37]. Previous reports indicated a possible role for ATP in controlling stomatal aperture[6,43], while other reports suggested a role for ATP in enhancing bacterial

pathogen resistance[15]. However, in neither case was a mechanism provided to explain these observations.

In order to examine these phenotypes further, we first examined the ability of ATP treatment to stimulate stomatal closure using treatment with abscisic acid (ABA) as a positive control (Fig. 5a, b). Addition of either ATP or the slowly hydrolyzed derivative ATPγS-triggered stomatal closure in the light while addition of ADP showed a similar but weaker ability. Similar assays conducted with either *dorn1-3* or *rbohd* mutant plants showed no closure of stomata upon addition of ATP, ATPγS, or ADP. However, these mutant plants did close stomata upon addition of ABA. We also tested the *dorn1-1* and *dorn1-2* mutant plants and also found that they did not close stomata in response to ATP treatment (Supplementary Fig. 5a). In addition,

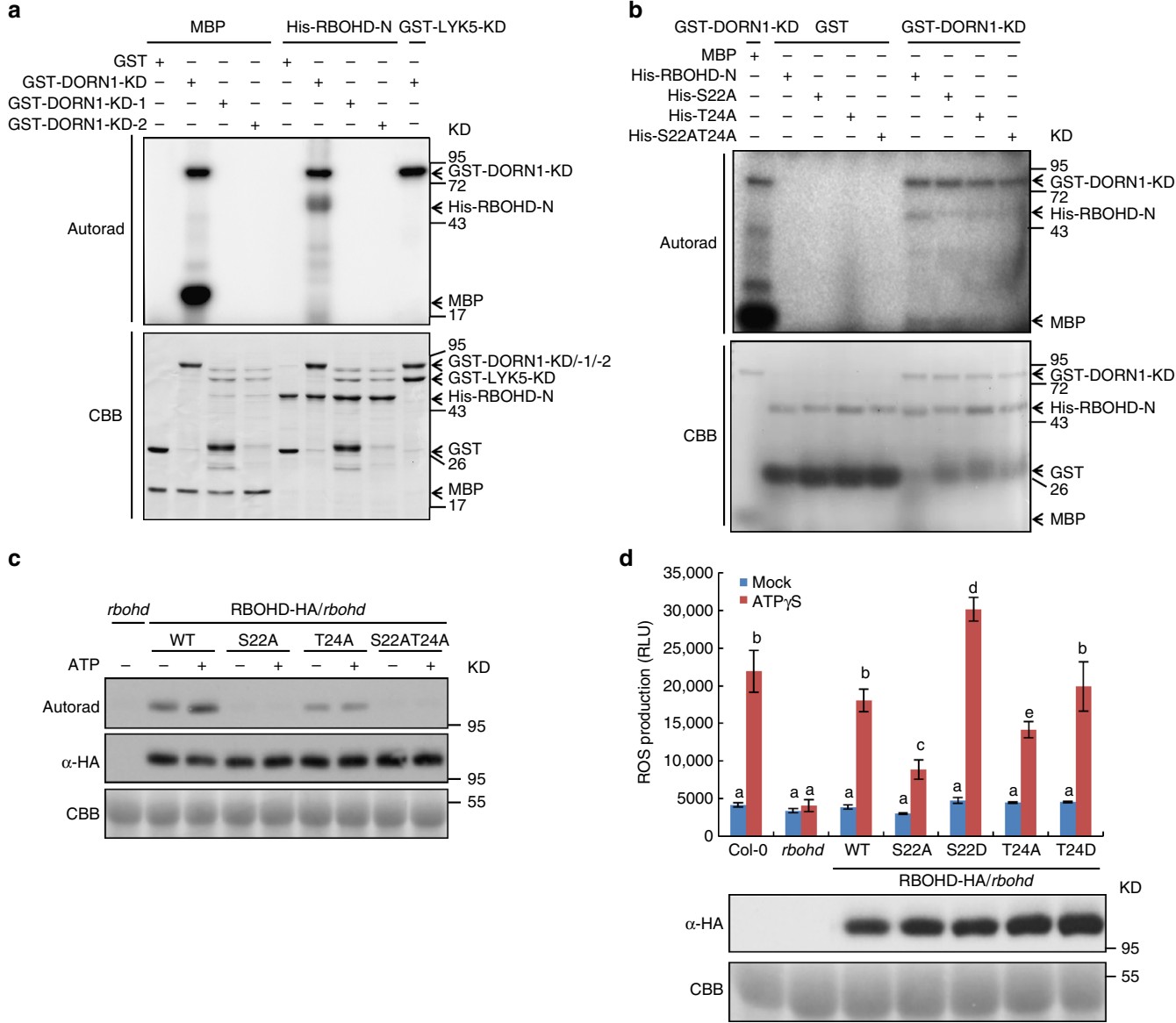

**Fig. 4** DORN1 phosphorylates RBOHD-N at S22 and T24 sites in vitro and in vivo. **a** DORN1 phosphorylates the N-terminal domain of RBOHD. Purified His-RBOHD-N recombinant protein was incubated with GST-DORN1-KD kinase domain, GST-DORN1-KD-1 (kinase dead), GST-DORN1-KD-2 (kinase dead), or GST in an in vitro kinase assay. Autophosphorylation and trans-phosphorylation were measured by incorporation of $\gamma$-[$^{32}$P]-ATP. MBP and GST-LYK5-KD kinase domain were used as positive and negative controls, respectively. The protein loading was measured by Coomassie brilliant blue (CBB) staining. This experiment was repeated three times with similar results. **b** DORN1 phosphorylates RBOHD-N at S22 and T24 sites in vitro. Purified GST or GST-DORN1-KD protein was incubated with His-RBOHD-N or the respective mutant proteins, S22A (His-S22A), T24A (His-T24A), S22AT24A (His-S22AT24A), followed by an in vitro kinase assay. This experiment was repeated three times with similar results. **c** ATP-induced phosphorylation of RBOHD through S22 and T24 sites in vivo. The indicated constructs pGBW14-RBOHD (WT, S22A, T24A, and S22AT24A) were transiently expressed in *rbohd* mutant protoplasts incubated with $\gamma$-[$^{32}$P]-ATP overnight. After treating with 200 μM ATP for 30 min, total protein was extracted and subjected to immunoprecipitation. Total RBOHD-HA protein was detected by anti-HA immunoblotting. Protein loading was monitored by CBB. This experiment was repeated three times with similar results. **d** RBOHD phosphosites are required for ATP-triggered ROS production. The indicated constructs were transiently expressed in *rbohd* mutant protoplasts and treated with or without 200 μM ATPγS. ROS production was measured after 30 min. RLU relative luminescence units; values represent the mean ± SEM, $n = 8$ (biological replicates). Means with different letters are significantly different ($P < 0.01$; one-sided ANOVA). Total RBOHD-HA protein was detected by anti-HA immunoblot and CBB staining was used to monitor protein loading. This experiment was repeated three times with similar results

transgenic *rbohd* mutant plants expressing the RBOHD wild-type protein were able to close stomata in response to ATPγS. Meanwhile, transgenic *rbohd* plants expressing the RBOHD S22A mutant protein, but not those expressing the T24A mutant protein, showed significantly greater stomatal aperture compared to wild-type plants (Fig. 5c). In the same experiment, *rbohd* plants expressing the RBOHD S22AT24A mutant protein were largely

insensitive to ATP treatment. These data show that ATP-induced stomatal closure acts through DORN1, as a result of activation of RBOHD activity through S22 phosphorylation. In addition, we also tested the effects of ATP addition on mutant plants defective in ABA recognition or response. Specifically, we found that *Arabidopsis ost1* (Open Stomata1 [OST1]) and *pyr1/pyl1/pyl2/pyl4* (PYRABACTIN RESISTANCE1 [PYR1], PYRABACTIN

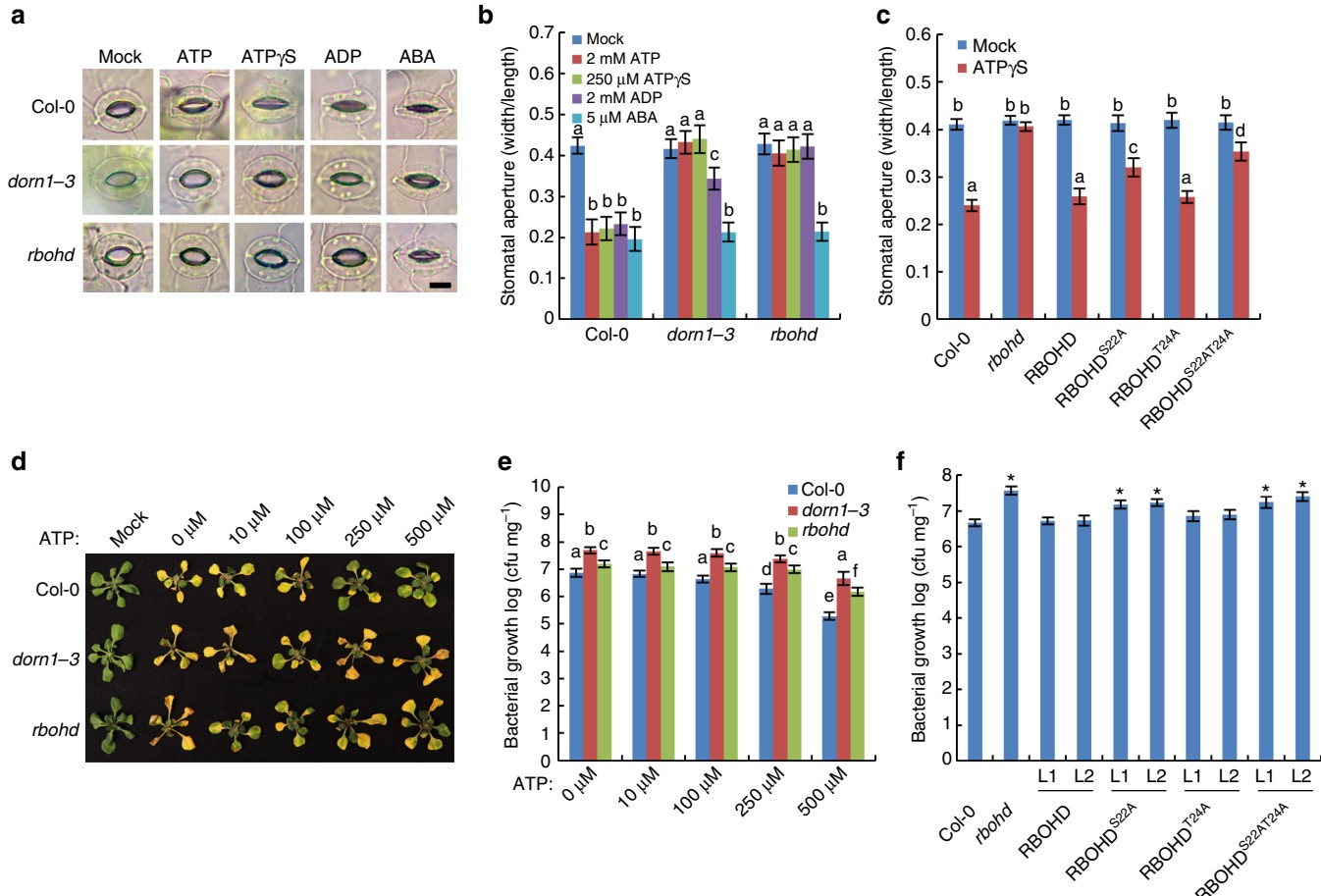

**Fig. 5** DORN1 and RBOHD positively regulate stomatal immunity. **a**, **b** DORN1 and RBOHD are required for the ATP- and ADP-induced stomatal closure. Stomatal aperture was measured after treatment with 2 mM ATP, 250 μM ATPγS, 2 mM ADP, or 5 μM ABA. Bar = 10 μm. Values represent the mean ± SEM, $n \geq 50$ (biological replicates); means with different letters are significantly different ($P < 0.01$; one-sided ANOVA). **c** DORN1-mediated RBOHD phosphosites are required for stomatal closure. The RBOHD transgenic lines in *rbohd* mutant background were used to measure stomatal closure after treatment with 250 μM ATPγS. Values represent the mean ± SEM, $n \geq 50$ (biological replicates); means with different letters are significantly different ($P < 0.01$; one-sided ANOVA). **d**, **e** DORN1 and RBOHD are required for stomatal immunity. Fourteen-day-old seedlings were flood inoculated with a *P. syringae* DC3000 suspension ($5 \times 10^6$ CFU ml$^{-1}$) with or without the addition of ATP. Bacterial colonization was determined by plate counting 3 days post inoculation. Values represent the mean ± SEM, $n \geq 8$ (biological replicates). Means with different letters are significantly different ($P < 0.05$; one-sided ANOVA). **f** DORN1-mediated RBOHD phosphosites are required for bacterial defense. Two independent T2 transgenic lines (L1 and L2) were used to measure bacterial growth after 3 days post inoculation with 200 μM ATP treatment. Values represent the mean ± SEM, $n \geq 8$ (biological replicates); *$P < 0.05$, Student's *t* test. All above experiments were repeated three times with similar results

RESISTANCE-LIKE1 [PYL1], PYL2, and PYL4) quadruple mutants, which do not close stomata upon ABA addition[44,45], were able to close stomata in response to ATP treatment (Supplementary Fig. 5b). Thus, this ATP-dependent pathway is independent of ABA-induced stomatal closure.

The observation that ATP treatment closes stomata through the action of DORN1 suggested an explanation for observations that ATP can enhance resistance to *P. syringae* DC3000. In order to examine this directly, we dip-inoculated plants with *P. syringae* DC3000 *COR⁻* bacteria, which are unable to produce the toxin coronatine (COR) that contributes to stomatal reopening during infection[26]. As expected, we observed stomatal closure with no reopening when plants were inoculated with the *P. syringae* DC3000 *COR⁻* bacteria. In contrast, the stomata of *dorn1-3* or *rbohd* mutant plants did not close upon bacterial inoculation (Supplementary Fig. 5c). Addition of ATPγS did not significantly increase stomatal closure when added along with bacterial inoculation, while addition of ABA resulted in a markedly stronger closure response. As expected, addition of ATPγS along with bacterial inoculation to either the *dorn1-3* or *rbohd* mutant

plants showed no effect on stomatal closure, while these plants remained sensitive to ABA addition (Supplementary Fig. 5c). The data show that bacterial recognition triggers stomatal closure through a mechanism that requires DORN1, likely mediated by DORN1 activation of RBOHD.

Direct, syringe-mediated infiltration of *P. syringae* into the leaf bypasses the need for entry through stomata. Therefore, in order to directly test the role of DORN1 in bacterial resistance, we used dip-inoculation to introduce the *P. syringae* pathogen, a method that requires entry through stomata[46]. The level of bacterial infection was monitored visually and direct counting was used to quantify the level of pathogen colonization (Fig. 5d, e). Relative to wild-type plants, bacterial colonization of the leaves of the *dorn1-3* and *rbohd* mutant plants was significantly greater 3 days post inoculation (dpi) (Fig. 5d, e). Furthermore, *rbohd* mutant plants expressing the RBOHD S22A or S22AT24A mutant proteins were significantly more susceptible than plants complemented with the wild-type RBOHD protein (Fig. 5f and Supplementary Fig. 5d). These data are consistent with the model by which bacterial recognition triggers ATP release, resulting in DORN1

phosphorylation of RBOHD on residue S22, ROS production and stomatal closure blocking entry of the bacterial pathogen. Interestingly, the *dorn1-3* and *rbohd* mutants exhibited gradually decreasing bacterial populations in an ATP dose-dependent manner (Fig. 5d, e), suggesting that there may be additional, unknown factors modulating ATP-mediated stomatal-bacterial immunity.

While we interpret our results as indicative of ATP-induced signaling, another possible explanation could be direct inhibition of bacterial growth by exogenous ATP. Therefore, to exclude this possibility, we added increasing levels of ATP to cultures of *P. syringae* DC3000 while monitoring growth by optical density. As shown in Supplementary Fig. 6, high levels (5–15 mM) of ATP did inhibit bacterial growth to some degree but no inhibition occurred with the lower, sub-mM concentrations used in our plant experiments.

Because water loss is important in stomatal immunity, we also investigated the transpiration activity of mutants and wild type in 4–5-week-old plant leaves. Leaves of wild-type plants lost significantly less weight than either *dorn1-3* or *rbohd* mutant plants due to water loss. In addition, decreased water loss was only found for wild-type plants when leaves were treated with 2 mM ATP, consistent with the observation that both *dorn1* and *rbohd* stomata remained open regardless of ATP treatment (Supplementary Fig. 7). Similar kinetics of water loss were observed when leaves were treated with *P. syringae* DC3000 especially at 3 h, suggesting that ATP and bacteria can regulate stomatal closure to prevent water loss during infection.

**Pathogen triggers ATP release and downstream signaling**. The data above are consistent with bacterial infection inducing the release of ATP. In order to test for this directly, we used light production by luciferase to assay for an increase in ATP on the leaf surface upon bacterial inoculation. As shown in Fig. 6a, b, a significant increase in extracellular ATP was detected using luciferase, especially around guard cells, in response to bacterial inoculation. However, the flg22-induced ATP release was lower than that seen upon *P. syringae* DC3000 inoculation, suggesting that the pathogen likely induces a stronger "stress" response than that of a single elicitor. These results are similar to those reported previously for guard cells[6] and also for root hair cells upon treatment with the fungal pathogen elicitor chitin[9].

We next performed a ROS burst assay in response to *P. syringae* inoculation and observed that ROS production was significantly reduced in *dorn1-3* mutant plants while undetectable in *rbohd* mutant plants (Fig. 6c). Interestingly, both ATP addition and *P. syringae* inoculation could induce DORN1 protein modification, likely due to protein phosphorylation, in *dorn1-3* mutant plants expressing a DORN1-3×HA fusion protein from the *DORN1* native promoter (Fig. 6d and Supplementary Fig. 8). However, the flg22-induced ROS burst in *dorn1-3* mutant and DORN1 protein phosphorylation was not affected.

**Discussion**
Both plant and animals recognize eATP as a DAMP to respond to stress or physical damage. Indeed, many of the cellular responses to eATP (e.g., ROS production) are similar between plants and animals. However, the receptors that mediated these responses are quite different. A great deal of research has gone into deciphering the cellular signaling pathways downstream of the mammalian P2Y and P2X purinergic receptors and these studies have led to the development of important drugs. In comparison, nothing is known of how ATP recognition by DORN1 is coupled to downstream signaling pathways. For example, although DORN1 was identified as the first plant eATP receptor, the

downstream substrates of DORN1 phosphorylation are unknown. However, clearly, the kinase activity of DORN1 is crucial for its role in purinergic signaling[7]. The key residues within the DORN1 kinase domain that are autophosphorylated in response to ATP and are essential for signaling are now identified as S391, S440, and S451. Mutation in any of these residues results in the production of a protein that is unable to complement the *dorn1* mutant phenotype.

Given that eATP is implicated in a variety of plant stress responses[8], it is likely that DORN1 signaling involves a variety of downstream target proteins. Therefore, RBOHD is likely only one of the substrates but its phosphorylation already explains some of the key physiological responses attributed to eATP. For example, the data show that DORN1 phosphorylation of RBOHD on residue S22 results in an activation of ROS production and stomatal closure providing enhanced resistance to bacterial pathogens that use stomata as their primary site of infection. Interesting parallels can be made with mammals. For example, the purinergic receptor P2X7 is known to activate ROS production via membrane-associated NADPH oxidase[47–49]. This can also provide protection against invading pathogens[47], as well as generally contributing to the inflammatory response. It is likely that these processes represent convergent evolution, where plants and animals have adapted purinergic signaling to provide for a robust stress response but using different receptors. An interesting, unanswered question is to what extent plant and animal purinergic pathways have been conserved.

The importance of ROS production and RBOHD is exemplified by the myriad ways in which the activity of this protein is regulated. For example, many stimuli, including eATP, trigger a rapid and strong increase in cytoplasmic calcium, which can then directly bind to RBOHD via the EF-hand motifs resulting in an increase in ROS production[50–52]. The response to calcium is also mediated through calcium-dependent protein kinases (CPKs) or calcineurin-B-like protein interacting protein kinases (CIPKs) that also regulate NADPH oxidase activity[34,35,53,54]. For example, CPKs were shown to phosphorylate RBOHD at residues Ser133, Ser148, Ser163, and Ser347[35]. In addition, PAMP treatment induces phosphorylation of the RBOHD residues Ser39, Ser148, Ser152, Ser163, Ser343, and Ser347[35,41,55]. In part, this is mediated by BIK1, which specifically phosphorylates RBOHD at residues Ser39, Ser339, Ser343, and Ser347[36,37]. Residue Ser347 is of particular importance since it can be phosphorylated by both BIK1 and CPKs[56].

Note that none of these identified kinases was shown to phosphorylate RBOHD residue S22, although a previous phosphoproteomic study showed that this residue was phosphorylated in planta[41]. We now identify DORN1 as the kinase that specifically targets S22 and by so doing activates NADPH oxidase activity. This adds a new element to the overall regulation of RBOHD allowing for integration of responses to PAMPs (via BIK1), DAMPs (via DORN1 and BIK1), and a variety of stresses (via calcium signaling). At this point, it remains unclear as to how or whether these various regulatory mechanisms interact to either antagonize or synergize the ROS response. The presence of these multiple regulatory pathways that control RBOHD activity may explain, in part, why *dorn1-3* mutant plants still retain some, residual ROS production.

ROS production acts as an important signal mediating both plant and animal immunity. One aspect of this in plants is the ability to induce closure of stomata, blocking a key path for pathogen entry[56–58]. The addition of ATP was previously shown to trigger ROS production[9,19]. Independent reports also indicated that ATP addition could induce closure of stomata[6,59]. However, in neither case was a mechanism provided for these phenomena. The data presented here now provide this mechanism by showing

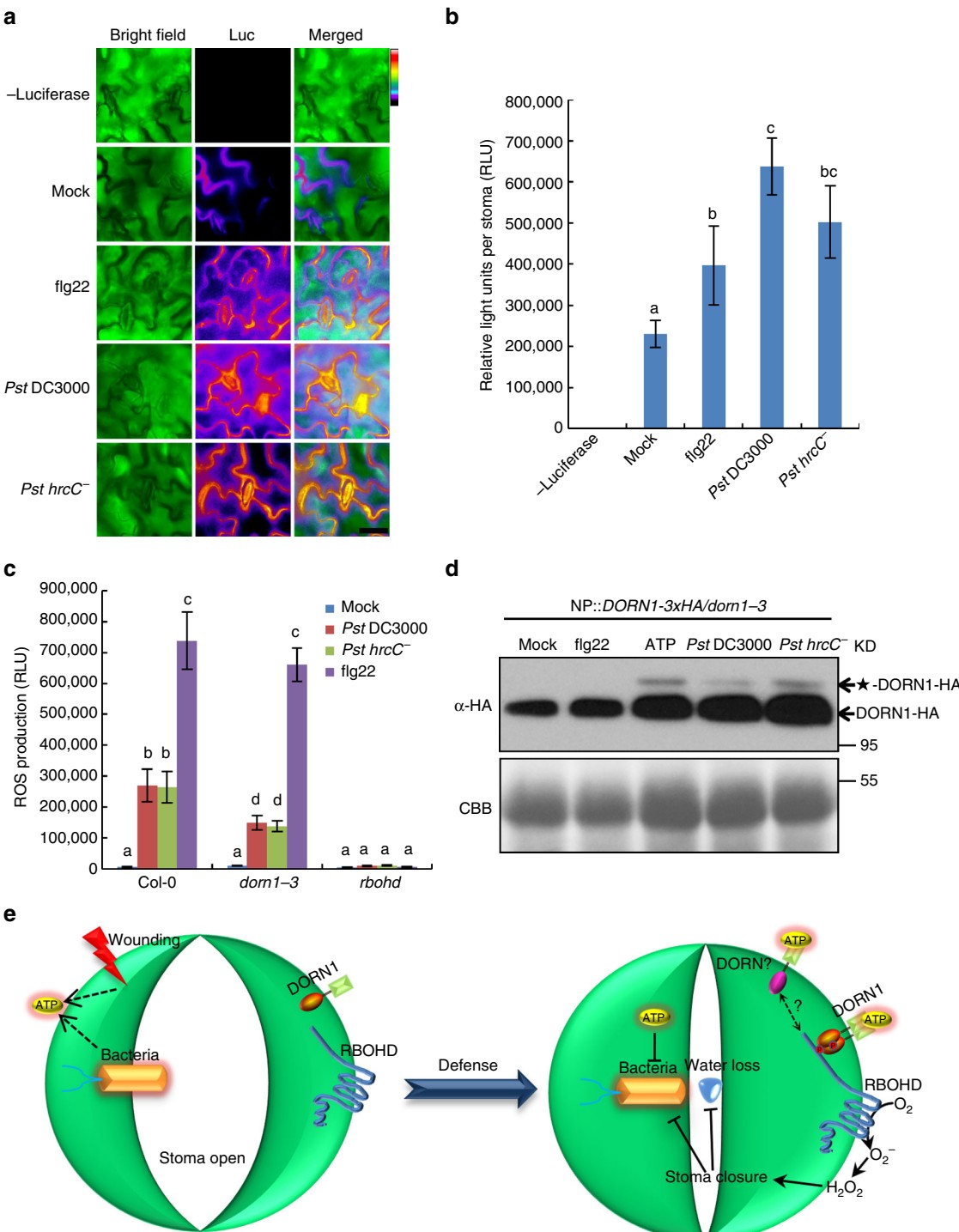

**Fig. 6** The release of ATP triggers DORN1 protein modification to regulate ROS during pathogen infection. **a** Bacteria induce ATP release in the guard cells. Luciferase fluorescence shows that ATP accumulates around the stomata on *P. syringae* DC3000, *hrcC*⁻ (OD₆₀₀ = 0.2) and 10 μM flg22 infecting plant leaves. Bar = 20 μm. **b** Relative quantification of the eATP concentration released from stomata. RLU relative light units; values represent the mean ± SD, *n* = 50 (biological replicates). Means with different letters are significantly different (*P* < 0.01; one-sided ANOVA). This experiment was repeated three times with similar results. **c** DORN1 and RBOHD are responsible for bacteria-induced ROS production. ROS production was measured using leaf discs inoculated with *P. syringae* DC3000, *hrcC*⁻ (OD₆₀₀ = 0.1) and 1 μM flg22. Values represent the mean ± SEM, *n* = 8 (biological replicates); means with different letters are significantly different (*P* < 0.01; one-sided ANOVA). This experiment was repeated three times with similar results. **d** Bacteria elicit DORN1 protein modification similar to ATP, likely due to protein phosphorylation. DORN1 protein modification (★-DORN1-HA) was determined in *NP:: DORN1-3 × HA/dorn1-3* expressing transgenic plants after treatment with 1 μM flg22, 250 μM ATP, *P. syringae* and *hrcC*⁻ (OD₆₀₀ = 0.2) for 30 min. CBB Coomassie brilliant blue staining. This experiment was repeated three times with similar results. **e** The proposed model depicting the mechanism by which ATP elicits DORN1-mediated RBOHD phosphorylation to regulate stomatal immunity

that ATP-induced stomatal closure depends on DORN1 and is mediated by direct phosphorylation of RBOHD. The addition of ATP to *dorn1* or *rbohd* mutant plants failed to close stomata, and this function was not restored by expression of RBOHD mutant proteins possessing the S22A modification. Interestingly, pathogen inoculation leads to a significant elevation of extracellular ATP, which appeared to be at its highest around the stomata guard cells. Therefore, in addition to inducing defense responses through PAMP-triggered immunity, the pathogen also is recognized by the plant through stress-induced release of eATP.

The stomata not only serve as a conduit for pathogen infection, but also function for gas and water exchange. Measurements of plant transpiration demonstrated that ATP can close the stomata to prevent plant water loss. Both *dorn1* and *rbohd* mutant plants, consistent with their open stomata, showed significantly higher transpiration loss of water. Thus, eATP, through the action of DORN1 and RBOHD, may be an important signal that helps mediate water homeostasis, a critical trait for crop production, especially under conditions of a changing climate.

In summary, the data support a model by which ATP, released as a result of stress (biotic or abiotic), is recognized by DORN1, triggering phosphorylation of the N terminus of RBOHD, resulting in elevated levels of ROS production that function to induce stomatal closure (Fig. 6e). These results are important since they clearly demonstrate an important role for purinergic signaling in plants, implicating eATP as a key signal in important physiological processes that impact photosynthesis, water homeostasis, pathogen resistance, and ultimately yield.

## Methods

**Plant materials and constructs**. All *Arabidopsis* plants used in this study utilized an aequorin-expressing transgenic line (Col-0 background), including *dorn1-3* (Salk_042209), *rbohd*[60], *dorn1-1*, and *dorn1-2*[7]. Plants were grown in soil or 1/2 MS medium containing 1% sucrose at 21–23 °C, 60–70% humidity and under long-day (16 h light/8 h dark) conditions.

Full-length *DORN1* (At5g60300) or *RBOHD* (At5g47910), as well as sub-genic fragments *DORN1-KD* (kinase domain) or *RBOHD-N* (N-terminal region, 1–1128 bp), were amplified using gene-specific primers (Supplementary Data 2) with cDNA derived from wild-type plants. The PCR products were cloned into pDONR-Zeo or pGEM-T Easy vectors. Full-length clones of the kinase-dead *dorn1-1* and *dorn1-2* mutants were amplified from genomic DNA. The *RBOHD*-S22A, S22D, T24A, T24D, and S22AT24A mutant clones were generated by site-directed mutagenesis.

The pUC-GW14 and pUC-GW17 destination vectors were used for LR cloning in order to express specific proteins in *Arabidopsis* protoplasts. For BiFC assays, the DNA from the pDONR-Zeo vector were cloned into pAM-PAT-35SS::YFP:GW, pAM-PAT-35SS::YFPc:GW, and pAM-PAT-35SS::YFPn:GW vectors[61] to form fusions with split YFP at the C-termini of proteins using LR cloning.

The DNA fragments of *DORN1-KD* and *RBOHD-N* cut with EcoRI and XhoI were inserted into pGEX-5X-1 and pET28a, respectively, to generate plasmids suitable for protein expression in *E. coli*.

In order to express specific proteins in transgenic plants, a DNA, promoter fragment representing 2041 bp 5′ of the *RBOHD* start codon was isolated by PCR from genomic DNA and then ligated with *RBOHD*, followed by cloning into the binary vectors pGBW13, pGBW14, and pGBW17[62] using LR cloning. In a similar fashion, the *DORN1* promoter was isolated from genomic DNA by PCR amplification of a 1.5 kb region 5′ of the start site, which was then fused with the *DORN1* gene mutant clones containing different autophosphorylation site forms that were generated by site-directed mutagenesis. Those combined fragments were cloned into the pGWB13 vector for plant transformation.

**Mapping of the autophosphorylation sites of DORN1**. Mapping of the DORN1 autophosphorylation sites was done using a bottom-up proteomics approach. The purified, recombinant DORN1 kinase was phosphorylated in vitro in the presence of 5 mM $MgCl_2$, $MnCl_2$, and/or both $MgCl_2$ and $MnCl_2$ with or without ATP. Proteins were subsequently digested in-solution with trypsin. The tryptic peptides were fragmented by using either collision-induced dissociation (CID) or by using a "decision tree" method, which utilizes both CID and ETD during a single sample analysis and analyzed by using a LTQ Orbitrap XL ETD mass spectrometer (Thermo Fisher, San Jose, CA). Acquired MS/MS spectra were searched against the entire *Arabidopsis* protein sequence database obtained from TAIR database (TAIR10) and concatenated to a randomized version of TAIR10 (i.e., decoy) generated using an in-house developed program (DecoyDB Creator, available at

www.oilseedproteomics.missouri.edu). Peptide spectral matches were evaluated primarily using the XCorr scoring function of SEQUEST employing a 1% false discovery rate. Phosphorylation site localization was performed using phosphoRS (Proteome Discoverer, version 1.0.3, Thermo Fisher). Each phosphopeptide spectrum was inspected manually and accepted only when the phosphopeptide had the highest pRS site probability, pRS score, XCorr value, and site-determining fragment ions allowed unambiguous localization of the phosphorylation site. This peptide score is based on the cumulative binomial probability that the observed match is a random event. The value of the pRS score strongly depends on the data scored and scores ≥50 was considered as a potential phosphopeptide. On the other hand, pRS site probability of each phosphorylation site is an estimation of the probability (0–100%) for the respective site being truly phosphorylated. pRS site probabilities above ≥95% are good evidence that the respective site is truly phosphorylated. The spectral counting method was employed to demonstrate the phosphorylation status in response to the given treatments. The mass spectrometry proteomics data have been deposited to the ProteomeXchange Consortium via the PRIDE partner repository with the data set identifier PXD006678.

**Calcium influx assays**. Briefly, 5-day-old seedlings were individually transferred to the wells of a 96-well plate with 50 μl of reconstitution buffer containing 10 μM coelenterazine (Nanolight Technology, Pinetop, AZ), 2 mM MES buffer (pH 5.7), and 10 mM $CaCl_2$ and incubated in the dark at room temperature overnight. Fifty microliters of treatment solution (concentration was double strength to give a set final concentration of 25 mM MES and 100 μM ATP (Sigma, A2383) was added to each well, and the luminescence was monitored using a CCD camera (Photek 216; Photek, Ltd.)[63].

**Kinase client assay (KiC assay)**. A library of more than 2100 peptides developed from identified phosphorylation sites taken from a number of studies was incubated with the purified, recombinant GST-DORN1-KD kinase domain in the presence of ATP. The peptide mixture was then analyzed by a Finnigan Surveyor liquid chromatography (LC) system attached to a LTQ Orbitrap XL ETD mass spectrometer. For analysis of KiC assay results, raw MS files were searched against a decoy database consisting of the random complement of the sequences comprising the peptide library, using SEQUEST (Proteome Discoverer, v. 1.0.3, Thermo Fisher). The instrument and detailed search parameters were used as before[38]. Identification data were evaluated using the XCorr function of SEQUEST, and phosphorylation site localization was accomplished using phosphoRS (Proteome Discoverer, v. 1.0.3, Thermo Fisher). The XCorr values for each charge state were set to default, and no decoy hits were allowed. For final validation, each spectrum was inspected manually and accepted only when the phosphopeptide had the highest pRS site probability, pRS score, XCorr value, and site-determining fragment ions allowed unambiguous localization of the phosphorylation site. Phosphopeptides with a pRS score ≥15 and/or a pRS site probability of ≥50% were accepted. Two sets of empty vectors (GST and MBP) and two kinase-dead proteins, GST-DORN1-1 (D572N) and GST-DORN1-2 (D525N), were used as negative controls.

**In vitro pull-down**. Recombinant proteins GST-DORN1-KD, GST-DORN1-KD-1, GST-DORN1-KD-2, and His-RBOHD-N (WT, S22A, T24A, and S22AT24A) were expressed in *E. coli* and affinity purified using the Glutathione Resin (GenScript) and TALON® Metal Affinity Resin (Clontech), respectively. For pull-down, 2 μg GST and His recombinant proteins were incubated with the pulldown buffer containing 50 mM Tris-HCl pH 7.5, 100 mM NaCl, and 0.5% Triton-X 100 for 2 h at 4 °C. After taking 20 μl as an input, 25 μl Glutathione Resin beads was added for 2 h and then washed seven times with the pulldown buffer. The protein was eluted by 25 μl 1× SDS-PAGE loading buffer and heated at 100 °C for 10 min. The proteins were separated using SDS-PAGE gels and detected by immunoblotting using anti-His (Cat no. SAB1305538, Sigma; dilution, 1:1000) and anti-GST-Hrp (Cat no. A01380-40, GeneScript, dilution, 1:1000).

**Arabidopsis protoplast isolation and transformation**. *Arabidopsis* protoplasts were isolated from leaf tissues of 14-day-old seedlings[64]. Approximately, 2 g of 14-day-old seedlings were sliced with a fresh razor blade into a 200 ml beaker with 15 ml of filter-sterilized TVL solution (0.3 M sorbitol and 50 mM $CaCl_2$). Next, 20 ml of filter-sterilized enzyme solution was added containing 0.5 M sucrose, 10 mM MES-KOH pH 5.7, 20 mM $CaCl_2$, 40 mM KCl, 1% cellulase (Onozuka R-10), 1% macerozyme (R-10), and covered with parafilm and aluminum foil. After a gentle swirling motion at room temperature for 16–18 h, the enzyme solution containing protoplasts were filtered through the 75-mm nylon mesh into a 50 ml tube. These protoplasts were gently covered with 10 ml W5 solution (2 mM MES pH 5.7, 154 mM NaCl, 125 mM $CaCl_2$, and 5 mM KCl) without disturbing the sugar solution gradient, followed by centrifugation for 7 min at $100 \times g$. The protoplasts in a 10 ml volume at the interface of enzyme solution and W5 solution were transferred to a new 50 ml tube. An aliquot of 15 ml W5 solution was added followed by centrifugation for 5 min at $60 \times g$. The protoplasts were then washed with 15 ml of W5 solution and centrifuged again for 5 min at $60 \times g$. Resuspend the pelleted protoplasts were resuspended in 1-3 ml MMG Solution containing 4 mM MES pH 5.7, 0.4 M mannitol and 15 mM $MgCl_2$. Plasmids encoding the proteins to

be expressed were transfected into protoplasts using the DNA-PEG-calcium transfection method[65]. Briefly, 10 µl DNA (10–20 µg of plasmid) was added to 100 µl protoplasts and mixed gently. An aliquot of 110 µl of PEG solution was mixed with DNA protoplasts by gently tapping the tube. This transfection mixture was incubated at room temperature for ~15 min. The transfection mixture was then diluted with 400–440 µl W5 solution to stop the transfection process followed by centrifugation at $100 \times g$ for 2 min at room temperature. The pelleted protoplasts were resuspended gently with 1 ml WI or W5 solution. After incubation overnight in the dark in a 23 °C growth chamber, 200 µM ATP was added to the protoplast solution for 20 min and used for the various assays.

**Co-immunoprecipitation assay.** Total protein was extracted from protoplasts (spun down) or plant tissues (ground in liquid nitrogen) with an extraction buffer containing 50 mM Tris (PH 7.5), 150 mM NaCl, 0.5% Triton-X 100, and 1× protease inhibitor (Sigma) for 30 min on ice. The solution was then centrifuged at $20,000 \times g$ for 15 min at 4 °C, the supernatant was decanted and 1 µg anti-Myc (Cat no. SAB4700447, Sigma) was added to the supernatant and incubated for 4 h or overnight with end-to-end shaking at 4 °C. Subsequently, 25 µl protein A resin was added for 2 h, spun down and washed seven times with washing buffer (50 mM Tris (pH 7.5), 150 mM NaCl). After washing, the resin were eluted with 25 µl 1× SDS-PAGE loading buffer and the eluent heated at 100 °C for 10 min. The proteins were separated by SDS-PAGE gel electrophoresis and detected by immunoblotting with anti-HA-HRP (Cat no. 12013819001, Roche; dilution, 1:1000), a horseradish peroxidase-conjugated antibody.

**Bimolecular fluorescence complementation assay.** Plasmids of N- and C-terminal YFP protein fusions were co-transformed into *Arabidopsis* protoplasts as described above and then incubated in a 23 °C growth chamber overnight in the dark. The YFP fluorescence was monitored using a Leica DM 5500B Compound Microscope with Leica DFC290 Color Digital Camera. FM4-64 (Invitrogen, T3166) was used as a counter stain for the plasma membrane marker.

**In vitro phosphorylation assays.** For the in vitro kinase assay, 2 µg of purified GST or GST-DORN1-KD kinase was incubated with 1 µg His-RBOHD-N (WT, S22A, T24A, and S22AT24A) as substrate in a 20 µl reaction buffer containing 50 mM Tris-HCl pH 7.5, 50 mM KCl, 10 mM MgCl2, 10 mM ATP, and 0.25 µl radioactive [γ-$^{32}$P] ATP for 30 min at 30 °C. The reaction was stopped by increasing the temperature to 100 °C for 5 min. After which 5 µl of 5× SDS loading buffer was added and the proteins were separated by electrophoresis in 12% SDS-PAGE gels, followed by autoradiography for 3 h. The proteins within the gel were visualized by staining with Coomassie blue. Myelin basic protein, GST, and GST-LYK5-KD were used as controls.

**In vivo phosphorylation assay.** The plasmids expressing RBOHD-3×HA, RBOHDS22A-3×HA, RBOHDT24A-3×HA, or RBOHDS22AT24A-3×HA were transfected into protoplasts derived from *rbohd* mutant seedlings followed by incubation overnight in the dark in W5 solution with 40 mCi ml$^{-1}$ $^{32}$P phosphate[66]. The protoplasts were then treated with 200 µM ATP and protein immunoprecipitation was performed as described above using agarose-conjugated anti-HA antibody (Cat no. A2095, Sigma).

**Oxidative (ROS) burst assay.** Leaf discs were taken from 5-week-old plants and incubated in the wells of a 96-well-plate with 50 µl ddH2O overnight. 50 µl 2× chemiluminescent luminol buffer was added to each well to be a final concentration of 25 µM luminol (L-012, Wako Chemicals USA Inc. Richmond, VA), 20 µg ml$^{-1}$ horseradish peroxidase (Sigma, P6782) and 250 µM ATPγS (Sigma, A1388) before measurement. Luminescence was immediately monitored using a CCD camera (Photek 216; Photek, Ltd.). In order to measure ROS production upon bacterial pathogen inoculation, we used log-phase wild-type *P. syringae* cultures diluted to an OD600 = 0.1[67]. ROS production in protoplasts was measured by resuspending the isolated protoplasts in assay buffer containing 10 mM MES pH 7.0, 1 mM CaCl2, and 0.1 mM KCl[42].

**Stomatal aperture measurement.** In order to ensure that the starting plants had fully open stomata, 4–5 weeks old plants were placed under light for 3 h. Leaf peels were obtained from the abaxial side and incubated in a buffer containing 10 mM MES pH 6.15, 10 mM KCl, and 10 µM CaCl2. After treated with 2 mM ATP, 250 µM ATPγS, 2 mM ADP (Sigma, A2754), 5 or 10 µM ABA (Sigma, A1049) or mock solution for 1 h, the samples were observed using a microscope (Nikon, Alphaphot2). The stomatal aperture was measured using ImageJ software.

**Transpiration measurement by water loss.** The leaves were excised from 4–5 weeks old plants and then immersed in buffer containing 10 mM MES pH 5.7, 10 mM MgCl2, 0.01% Silwet-77, and 2 mM ATP or buffer containing *P. syringae* at a final OD600 = 0.01. Subsequently, samples were exposed to a partial vacuum for 5 min, blotted dry on a paper towel, and individually weighed immediately. The leaves were incubated on the bench top under light in a Petri plate with the cover left on and weighed at different times to follow water loss under drying conditions.

**qRT-PCR.** Leaf discs were taken from 5-week-old plants and incubated in sterile ddH2O at room temperature overnight. Samples were collected after treatment with 250 µM ATP, and then total RNA was extracted using Trizol reagent (Invitrogen) according to the manufacturer's instructions. The RNA concentration was estimated followed by treatment with Turbo DNA-free DNase (Ambion). The RNA was then used for first-strand cDNA synthesis using reverse transcriptase (Promega). The real-time PCR was performed using the iTaq™ Universal SYBR® Green Supermix (Bio-Rad) following the manufacturer's instructions. The gene-specific primers used are listed in the Supplementary Data 2. RNA levels were normalized against the expression of the *UBIQUITIN* (*UBQ*) gene.

**Bacterial growth assays.** Bacterial growth was performed using flood inoculation of seedlings assay[46]. Generally, 50 ml of *P. syringae* pv. *tomato* DC3000 ($5 \times 10^6$ colony-forming units (CFU) ml$^{-1}$) bacterial suspension containing 10 mM MES pH 5.7, 10 mM MgCl2, 0.025% Silwet L-77 with or without added ATP was dispensed into plates containing 2-week-old seedlings for 2–3 min at room temperature. The bacterial suspension was then immediately removed by decantation and the plates containing inoculated plants were incubated in the light in a 23 °C growth chamber. After 3 days post inoculation, the roots of seedlings were removed and the seedlings were washed with ddH2O for 5 min. The seedlings without roots were ground in 10 mM MgCl2, diluted serially, and plated on LB agar with 25 mM rifampicin. Colonies (CFU) were counted after incubation at 28 °C for 2 days.

**ATP release assay.** Plant leaves were incubated with 100 µg ml$^{-1}$ luciferase (Sigma, SRE0045) protein solution for 1 h at room temperature, followed by addition of a *P. syringae* suspension (OD600 = 0.2), which was then incubated for 30 min in 23 °C under light before washing three times with sterile ddH2O. Subsequently, samples were treated with a flash assay buffer containing 20 mM Tricine, 3 mM MgSO4, 0.1 mM EDTA, 2 mM dithiothreitol, and 500 µM D-luciferin (Sigma, L6882), and fluorescence was monitored using a Zeiss Axiovert 200 M with Leica DFC290 color camera and DFC340 monochrome camera.

**Data availability.** The mass spectrometry proteomics data are available at the ProteomeXchange Consortium via the PRIDE partner repository with the data set identifier PXD006678. The authors declare that all other data supporting the findings of this study are available within the manuscript and its supplementary files or are available from the corresponding author on request.

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

## Acknowledgements

Special thanks to Drs Jeongmin Choi and Kiwamu Tanaka who helped to initiate this project. We thank Professor Julian Schroeder for providing the *ost1* and *pyr1/pyl1/pyl2/pyl4* quadruple mutants. Research reported in this publication was supported by the National Institute of General Medical Sciences of the National Institutes of Health (grant no. R01GM121445 to G.S.). The content is solely the responsibility of the authors and does not necessarily represent the official views of the National Institutes of Health. This work was also supported by the next-generation BioGreen 21 Program Systems and Synthetic Agrobiotech Center, Rural Development Administration, Republic of Korea (grant no. PJ01116604 to G.S.) and by the Fundamental Research Funds for the Central Universities (No. 2662015PY165 to Y.C.).

## Author contributions

G.S., J.T., and D.C. designed the experiments. D.C., Y.C., H.L., and D.K. performed the experiments. N.A. performed the mass spectrometry analysis. G.S. and D.C. wrote the manuscript. All authors discussed the results and commented on the manuscript.

## Additional information

**Competing interests:** The authors declare no competing financial interests.

