## [Peer Review File · Nature Communications]

Reviewers' comments:

Reviewer #1 (Remarks to the Author):

The manuscript "Extracellular ATP elicits DORN1-mediated RBOHD phosphorylation to regulate stomatal aperture" by Chen et al. describes an area of research of great interest for the study of plant stress response. Focused on the role of ATP recognition by the receptor DORN1, the mass spectrometric-based approaches described herein have the potential to help understand the role of ATP as an extracellular signal in plants. While the significance behind this work is clear and reflected in the text, additional elaboration of the MS data would facilitate acceptance of stated conclusions.

General comments:

- Making raw MS data publicly available through a repository such as PRIDE would allow for a more thorough review of the stated conclusions.
- Supplemental data tables highlighting the phosphosite scoring/confidence in addition to quantitative mass spectrometric metrics (PSMs, replicates, scores etc...) would be useful when determining the significance of the results .

Results:

DORN1 autophosphorylation is essential for downstream signaling.

- Are the 13 sites all novel?
- What is the localization criterion for each of these sites. For example, the S709, 712,713,715,716 are all on the same tryptic peptide (MVTLP AEDPQSNHSSISSQR). Is there individual spectra for each of these? Confirmed in another way? PhosphoRS was described for later experiments, was it used here too with info like described in Fig 2a.
- With respect to Fig. 1a, the highlighted example is for an autophosphorylation site from a phosphopeptide containing one possible phosphorylation site. A better example might be to show an MS/MS spectra from a peptide with multiple S/T sites.
- When describing quantitative MS, the methods and supplemental information do not give enough detail to adequately assess the significance. Supplemental figure 1b mentions PSMs but this is not mentioned in the text along with the software and post-acquisition metrics used.
- In supplemental figure 1b, the y-axis says "Number of PMS" when it should say "Number of PSMs"

Kinase Client assay (KiC assay).

- For PhosphoRS scoring, a threshold of >50% accepted seems rather low to qualify as "allowed unambiguous localization of the phosphorylation site."

Figure 1

- Is there any information on location of binding of RBOHD within the kinase domain?

Reviewer #2 (Remarks to the Author):

This study establishes a connection from the extracellular ATP receptor kinase, DORN1, to the reactive burst enzyme, RBOHD, in Arabidopsis and establishes it as an important pathway mediating stomatal closure associated with plant defense against surface-inoculated *P. syringae*. DORN1 in vitro autophosphorylation sites are identified and phospho-null and phospho-mimic derivatives of three of these sites are shown to be important in vivo for DORN1 induction of Ca⁺ release in response to ATP (Fig. 1). DORN1 is shown to interact with RBOHD in vitro and in an ATP-enhanced manner in protoplasts and in Arabidopsis plants, and DORN1 contributes significantly to the RBOHD-dependent

ROS production in response to ATP (Fig. 2 and 3). DORN1 is shown to phosphorylate RBOHD in vitro and in protoplasts on S22 (and to a lesser extent on T24) and phospho-null and phospho-mimic derivatives of S22 in RBOHD support reduced and increased ROS production, respectively, in response to ATP when expressed in rbohd protoplasts (Fig. 4). DORN1 and RBOHD, dependent on residue S22, are shown to be required for stomatal closure in response to ATP and DC3000 COR-, but not ABA (Fig. 5 and S3). DORN1 and RBOHD, dependent on residue S22, are shown to contribute to resistance of surface inoculated DC3000 (Fig. 5). Finally, surface inoculation of DC3000 is shown to stimulate release of extracellular ATP, ROS production that is partially DORN1- and entirely RBOH-dependent, and phosphorylation of RBOHD (Fig. 6). This is an excellent piece of work that makes a mechanistic connection and links it to a biologically significant output. While I am enthusiastic about the work, I do have several comments to hopefully help strengthen it.

1. To confirm the specificity of the S22 phospho-null and phospho-mimic derivatives of RBOHD to DORN1 activation, these derivatives should be assessed for their ability to trigger ROS burst in response to stimuli that trigger DORN1-independent activation of RBOHD. Ideally this would include assays for both BIK1- and calcium-dependent activation.
2. For the DORN1 autophosphorylation site derivatives, fig. 1b-d, the authors refer to immunoprecipitation "data not shown" demonstrating similar protein expression. Data demonstrating that (from transgenic plants, not protoplasts, as in fig. 1e) should be included.
3. The derivatives of autophosphorylation site other than S391, S440, and S451, are asserted to provide no evidence that those residues play a role in DORN1-mediated ATP signaling. Those data should be included as a supplemental figure.
4. The full list of peptides identified in the kinase client assay should be shown.
5. For the Co-IP of DORN1 and RBOHD from Arabidopsis protoplasts, the CERK1 kinase domain is used as a control. To control for non-specificity of a Co-IP of membrane localized proteins, full-length CERK1 (or some other plasma membrane localized protein) should be used for this control.
6. The authors favor a model of a direct link from DORN1 protein to RBOHD protein (and do have significant data to support the model). However, the much stronger transcriptional induction of RBOHD by ATP in Col-0 relative to dorn1-3 from 15 minutes to 1 hour (Fig. S2), also supports an alternate (and non-mutually exclusive) hypothesis that the DORN1-dependent induction of ROS by ATP from 0 to 30 minutes (Fig. 2c), is a result of lower transcriptional induction of the gene. On line 163, the authors should mention this "indirect" hypothesis in addition to the "direct" hypothesis.
7. The text (lines 261-263) indicates that growth of surface-inoculated DC3000 was significantly greater on dorn1-3 and rbohd relative to Col-0. However, the ANOVA for figure 5e do not support this statement for rbohd. Is that statement based on alternate statistical testing? If not, it should be corrected.
8. Figure 5f should include the rbohd mutant as a control. Are the infection assays of figure 5f carried out by the same method as figure 5e? Are they carried out in the absence of supplemental ATP? Assuming the answer to both questions is yes, then it is important to explain why the S22A and S22AT24A derivatives support increased bacterial growth that is not observed with the rbohd mutant in figure 1e.
9. The rapid induction by DC3000 of extracellular ATP, DORN1-dependent induction of ROS, and DORN1 phosphorylation by DC3000 (Fig. 6) are each rapid responses, indicating that one or all may be PAMP-induced. The authors should repeat the experiments of Figure 6 with inclusion of the hrc- derivative of DC3000, flg22 and elf18.

Minor edits:

- The sentence from line 70-72 seems to belong in the preceding paragraph.
- Line 121 should read "DORN1-mediated" rather than "DORN1-mediate".
- Perhaps it is better call it DORN1 self-association rather than homodimer formation, as there is no evidence distinguishing between dimers or higher order complexes.

- On line 266, the reference to supplementary figure 3b does not pertain to the sentence.
- Could the authors speculate on why ATP promotes stomatal closure in the light, but opening in the dark?

Reviewer #3 (Remarks to the Author):

Extracellular ATP (eATP) is among the characterized DAMPs that can elicit immune responses in plants, and the LecRK DORN1 was previously identified as the Arabidopsis eATP receptor. In this manuscript, Chen et al present evidence that DORN1 can directly phosphorylate the NADPH oxidase RBOHD, which is responsible for producing the ROS burst that is characteristic of PTI signaling in response to many PAMPs/DAMPs. These findings are of great significance, as they would provide the first evidence for the direct trans-phosphorylation of RBOHD by a pattern recognition receptor (PRR). While the manuscript is well-written and the evidence general supports their main findings, the model presented by the authors is not entirely comprehensive and may require revision. These comments along with others are outlined below. Such issues aside, this manuscript provides a valuable contribution to our understanding of PTI signaling.

The model presented in Figure 6 seems overly simplified, as it suggests that direct transphosphorylation of RBOHD by DORN1 is sufficient for the complete activation of RBOHD downstream of DAMP (eATP) perception, however, this is not supported by the authors own data (Fig 5c). Also, since it was previously shown that the cytoplasmic kinase BIK1 phosphorylates RBOHD downstream of the Atpep receptor (PEPR1) and PAMP receptors (i.e. FLS2, EFR) (Kadota et al., Mol Cell 2014; Li et al., Cell Host Microbe 2014), it is essential that the authors test whether BIK1 is required for eATP-induced ROS production (as well as eATP-induced stomatal closure), and similarly whether mutations in BIK1-regulated RBOHD phosphosites affect these responses.

Furthermore, given these previous results and the fact that Atpep is a DAMP, it is an overstatement to suggest that PAMP perception activates RBOHD via BIK1, while DAMPs do so directly via their respective PRRs, as the authors currently suggest in their discussion.

Also, the fact that some eATP-induced responses are still observed in *dorn1-3* directly challenges the fact that DORN1 is the eATP receptor (Choi et al., Science 2014). It is not correct to state that the "presence of...multiple regulatory pathways that control RBOHD actively liklet explains why *dorn1-3* mutant plants still retain some, residual ROS production" (lines 347-349). If DORN1 is indeed the eATP receptor, then, all regulatory pathways and downstream signals triggered in response to eATP treatment should be abolished (if *dorn1-3* is a null mutant or a complete loss-of-function mutant). Based on these comments, it is also not correct to state in the legend of Figure 2 that DORN1 is "essential for the ATP-induced ROS burst" (lines 772-773) given the substantial ATP-induced ROS burst in *dorn1-3*.

Appropriate negative controls for BiFC assays are lacking, as it is not surprising that an unfused YFPc does not reconstitute a BiFC signal when expressed with a protein-YFPc fusion. See this recent commentary about appropriate controls for BiFC experiments:
<http://www.plantcell.org/content/28/5/1002.long>.

Minor comments:

- Lines 92-98: the authors only provide a partial explanation to the observations made previously by Clark et al. (2011), as none of their data explains why eATP stimulates stomatal opening in darkness. Please rephrase accordingly.
- Fig 1e: the IP sample should be shown along with coIP to show relative enrichment

- Line 151: it is not clear with the authors only used the kinase domain of CERK1 here, while full-length DORN1 was otherwise used.
- Fig 3b: the FM64 PM marker in the YFP control clearly co-localizes with cytosolic YFP signal, which suggests the signal is not specific.
- Fig 4b: The evidence that RBOHD-N transphosphorylation is different on the three S/T A mutants is not convincing - was this experiment repeated?
- Line 242: where are the data showing that the "ATP-dependent pathway is independent of ABA-induced stomatal closure? This hasn't been directly tested.
- Fig 6b: "Quantification of eATP concentration" is not accurate - the RLU value does allow for the calculation of concentration.
- Fig S3b: this suggests that no other PAMPs/DAMPs modulate stomatal aperture, as dorn1-3 mutants are nearly completely insensitive to DC3000 cor- - what about flagellin, etc, which have been previously shown to induce stomatal closure during infection (Zeng & He, Plant Physiol 2010)?
- Fig S3c: the different constructs are clearly expressed at different levels

Reviewers' comments:

Reviewer #1 (Remarks to the Author):

The manuscript “Extracellular ATP elicits DORN1-mediated RBOHD phosphorylation to regulate stomatal aperture” by Chen et al. describes an area of research of great interest for the study of plant stress response. Focused on the role of ATP recognition by the receptor DORN1, the mass spectrometric-based approaches described herein have the potential to help understand the role of ATP as an extracellular signal in plants. While the significance behind this work is clear and reflected in the text, additional elaboration of the MS data would facilitate acceptance of stated conclusions.

General comments:

- Making raw MS data publicly available through a repository such as PRIDE would allow for a more thorough review of the stated conclusions.

Answer: We appreciate the reviewer’s valuable suggestion. As suggested, all MS RAW data files (a total of 76 files) are now deposited to PRIDE. The following statement is now included in the revised method section.

“The mass spectrometry proteomics data have been deposited to the ProteomeXchange Consortium via the PRIDE partner repository with the dataset identifier PXD006678”.

- Supplemental data tables highlighting the phosphosite scoring/confidence in addition to quantitative mass spectrometric metrics (PSMs, replicates, scores etc…) would be useful when determining the significance of the results.

Answer: As suggested, we now provide Supplementary Data 1, which includes all the details related to phosphopeptide identification.

Results:

DORN1 autophosphorylation is essential for downstream signaling.

- Are the 13 sites all novel?

Answer: We checked the specificity in plants using the website: <http://musite.net>, which showed that most of them had more than 50% specificity in green plants. See the below table.

Position	Amino Acid	Surr. Sequence	Score	Specificity
341	T	AHRFSYRSLFKATKGFSKDEFLGKG	-1.32	59.03%
345	S	SYRSLFKATKGFSKDEFLGKGGFGE	-0.94	81.15%
391	S	DEGVKQFVAEVVSMRCLKHRNLVPL	-1.42	49.21%
440	S	HLFDDQKPVLSWSQRLVVVKGIASA	-1.48	43.62%
451	S	WSQRLVVVKGIASALWYLHTGADQV	-1.42	49.76%
458	T	VKGIASALWYLHTGADQVVLHRDVK	-1.37	54.70%
472	S	ADQVVLHRDVKASNIMLDAEFHGRL	-1.77	17.07%
683	S	DQPWGQTIDTKNSLHIVAEPEKPS	-1.68	24.38%
709	S	VKMVTLPAEDPQSNHSSISSR***	-1.36	55.47%
712	S	VTLPAEDPQSNHSSISSR*****	-1.28	62.09%
713	S	TLPALPAEDPQSNHSSISSR*****	-1.93	8.87%

715	S	PAEDPQSNHSSISSQR*****	-1.1	73.87%
716	S	AEDPQSNHSSISSQR*****	-1.13	72.15%

• What is the localization criterion for each of these sites. For example, the S709, 712,713,715,716 are all on the same tryptic peptide (MVTLP AEDPQSNHSSISSQR). Are there individual spectra for each of these? Confirmed in another way? PhosphoRS was described for later experiments, was it used here too with info like described in Fig 2a.

Answer: Thanks for asking this valuable question. PhosphoRS, a built-in parameter in the Proteome Discover MS/MS data search engine software, was used for identification and phosphosite localization for peptides identified in this study.

Yes, for that specific tryptic peptide (MVTLP AEDPQSNHSSISSQR) we identified individual spectra for each phosphosite. Brief information on each DORN1 autophosphorylation site is given below. For details, please see the newly added Supplementary Table 1.

Table 1. *In vitro* autophosphorylated sites of *Arabidopsis* DORN1.

Phosphopeptide	Position	pRS Score	pRS site probability	MH+ [Da]	XCorr
GFSKDEFLGK	S345	77	S(3): 100.0	1207.54302	2.49
QFVAEVVSMR	S391	144	S(8): 100.0	1245.56902	2.81
ELLLVSEYMPNGSLDEHLFDDQKPV LSWSQR	S440	26	S(13): 0.5; S(27): 20.6; S(29): 78.0	3725.76670	4.92
GIASALWYLHTGADQVVLHR	S451	112	S(4): 99.5; Y(8): 0.5; T(11): 0.0	2287.13970	5.60
DVKASNIMLDAEFHGR	S472	130	S(5): 100.0	1882.85808	5.01
NSLHIVAEPEKPSPAVK	S683	91	S(2): 100.0; S(13): 0.0	1895.97063	3.41
MVTLP AEDPQSNHSSISSQR	S709	67	T(3): 0.0; S(11): 76.7; S(14): 0.3; S(15): 10.7	2280.00371	2.67
MVTLP AEDPQSNHSSISSQR	S712	103	S(14): 92.1; S(15): 6.4; S(17): 0.5	2264.00859	4.48
MVTLP AEDPQSNHSSISSQR	S713	90	S(15): 85.6; S(17): 0.6; S(18): 0.1	2264.00591	4.55
MVTLP AEDPQSNHSSISSQR	S715	80	S(15): 8.7; S(17): 82.4; S(18): 8.7	2264.01103	3.69
MVTLP AEDPQSNHSSISSQR	S716	36	S(14): 83.4; S(15): 83.4; S(17): 38.1; S(18): 86.1	2423.94219	2.15

ATKGFSKDEFLGK	T341	47	T(2): 100.0; S(6): 100.0	1587.68816	2.23
GIASALWYLHTGADQVVLHR	T458	116	S(4): 0.0; Y(8): 0.0; T(11): 100.0	2287.13896	5.45

- With respect to Fig. 1a, the highlighted example is for an autophosphorylation site from a phosphopeptide containing one possible phosphorylation site. A better example might be to show an MS/MS spectra from a peptide with multiple S/T sites.

Answer: Like most labs, we include a novel phosphorylation assignment with good fragmentation as an example for a figure. Multisite phosphorylation events typically do not fly or fragment as well as mono site events, and thus we rely on current assignment standards for such peptides rather than de novo sequencing. Our multisite phosphorylation data met these requirements and was validated in some instances as illustrated here. We feel this is sufficient.

- When describing quantitative MS, the methods and supplemental information do not give enough detail to adequately assess the significance. Supplemental figure 1b mentions PSMs but this is not mentioned in the text along with the software and post-acquisition metrics used.

Answer: We appreciate that the reviewer pointing out this issue. Yes, the reviewer is right that Spectral counting was used to demonstrate the autophosphorylation efficiency and/or effect in response to the different conditions used in this study. We have revised the text to clarify this point. Please see in the Methods (under the subheading “Mapping of the autophosphorylation sites of DORN1”) and results section.

- In supplemental figure 1b, the y-axis says “Number of PMS” when it should say “Number of PSMs”

Answer: We apologize for this careless typo. This is now corrected in the revised supplemental figure 1b. Thanks.

Kinase Client assay (KiC assay).

- For PhosphoRS scoring, a threshold of >50% accepted seems rather low to qualify as “allowed unambiguous localization of the phosphorylation site.”

Answer: Thanks for this comment. Normally we would agree with this statement when one is performing true discovery phosphoproteomics with a highly complex sample (>1 M peptides) that is fraught with false-positives. However, with the KiC assay the sample complexity is considerably lower (200 peptides/assay). We verified that we can lower the PhosphoRS score under such conditions using known kinase-client relationships, and have published this finding.

See: “Ahsan, N. et al. A versatile mass spectrometry-based method to both identify kinase client-relationships and characterize signaling network topology. *J. Proteome Res.* 12, 937-948, (2013).” and “Brauer, E. K. et al. The Raf-like Kinase ILK1 and the High Affinity K⁺ Transporter HAK5 Are Required for Innate Immunity and Abiotic Stress Response. *Plant Physiol.* 171, 1470-1484, (2016).”

Figure 1

- Is there any information on location of binding of RBOHD within the kinase domain?

Answer: No...this is not a question that we have addressed. Although, admittedly of some interest,

answering this question would add little to the overall message of the current paper.

Reviewer #2 (Remarks to the Author):

This study establishes a connection from the extracellular ATP receptor kinase, DORN1, to the reactive burst enzyme, RBOHD, in Arabidopsis and establishes it as an important pathway mediating stomatal closure associated with plant defense against surface-inoculated *P. syringae*. DORN1 in vitro autophosphorylation sites are identified and phospho-null and phospho-mimic derivatives of three of these sites are shown to be important in vivo for DORN1 induction of Ca⁺ release in response to ATP (Fig. 1). DORN1 is shown to interact with RBOHD in vitro and in an ATP-enhanced manner in protoplasts and in Arabidopsis plants, and DORN1 contributes significantly to the RBOHD-dependent ROS production in response to ATP (Fig. 2 and 3). DORN1 is shown to phosphorylate RBOHD in vitro and in protoplasts on S22 (and to a lesser extent on T24) and phospho-null and phospho-mimic derivatives of S22 in RBOHD support reduced and increased ROS production, respectively, in response to ATP when expressed in *rboh*d protoplasts (Fig. 4). DORN1 and RBOHD, dependent on residue S22, are shown to be required for stomatal closure in response to ATP and DC3000 COR-, but not ABA (Fig. 5 and S3). DORN1 and RBOHD, dependent on residue S22, are shown to contribute to resistance of surface inoculated DC3000 (Fig. 5). Finally, surface inoculation of DC3000 is shown to stimulate release of extracellular ATP, ROS production that is partially DORN1- and entirely RBOH-dependent, and phosphorylation of RBOHD (Fig. 6). This is an excellent piece of work that makes a mechanistic connection and links it to a biologically significant output. While I am enthusiastic about the work, I do have several comments to hopefully help strengthen it.

1. To confirm the specificity of the S22 phospho-null and phospho-mimic derivatives of RBOHD to DORN1 activation, these derivatives should be assessed for their ability to trigger ROS burst in response to stimuli that trigger DORN1-independent activation of RBOHD. Ideally this would include assays for both BIK1- and calcium-dependent activation.

Answer: We appreciate that the reviewer pointing out this issue. We carried out *flg22* and chitin-elicited ROS burst in S22A and S22D expressing transgenic plants (Supplementary fig 4, 5c), the ROS burst was not significantly different between wild type and the S22 phospho-null and phospho-mimic, which was also described in Nuhse et al., 2007. So S22 is specific for ATP-DORN1 activation of RBOHD. Because the *rboh*d mutant does not have a calcium influx phenotype, it is difficult to investigate S22 specific effects using the *rboh*d mutant.

See: “Nuhse, T. S., Bottrill, A. R., Jones, A. M. & Peck, S. C. Quantitative phosphoproteomic analysis of plasma membrane proteins reveals regulatory mechanisms of plant innate immune responses. *Plant J.* 51, 931-940, (2007).”

2. For the DORN1 autophosphorylation site derivatives, fig. 1b-d, the authors refer to immunoprecipitation “data not shown” demonstrating similar protein expression. Data demonstrating that (from transgenic plants, not protoplasts, as in fig. 1e) should be included.

Answer: Thanks for asking this valuable question. We show these data in Supplementary fig. 2a.

3. The derivatives of autophosphorylation site other than S391, S440, and S451, are asserted to provide no

evidence that those residues play a role in DORN1-mediated ATP signaling. Those data should be included as a supplemental figure.

Answer: Thanks for asking this valuable question. We show these data in Supplementary fig. 2b.

4. The full list of peptides identified in the kinase client assay should be shown.

Answer: Thanks for asking this valuable question. Although we understand the curiosity of the reviewer, adding this list would add nothing to the general message of this paper, which is focused specifically on the role of DORN1 in relation to RBOHD. Each of the proteins identified by the KiC assay represents a separate project and several of these are now under investigation in the laboratory and will be the subject of subsequent publications. Indeed, since the list clearly contains some false positives, one could argue that releasing the invalidated list would actually mislead readers. We prefer to do the additional work needed to confirm each of the putative DORN1 partners and publish these as each story is completed.

5. For the Co-IP of DORN1 and RBOHD from Arabidopsis protoplasts, the CERK1 kinase domain is used as a control. To control for non-specificity of a Co-IP of membrane localized proteins, full-length CERK1 (or some other plasma membrane localized protein) should be used for this control.

Answer: We apologize for this careless typo. In this Co-IP, we used the full-length CERK1 and this is now clarified in the text.

6. The authors favor a model of a direct link from DORN1 protein to RBOHD protein (and do have significant data to support the model). However, the much stronger transcriptional induction of RBOHD by ATP in Col-0 relative to *dorn1-3* from 15 minutes to 1 hour (Fig. S2), also supports an alternate (and non-mutually exclusive) hypothesis that the DORN1-dependent induction of ROS by ATP from 0 to 30 minutes (Fig. 2c), is a result of lower transcriptional induction of the gene. On line 163, the authors should mention this “indirect” hypothesis in addition to the “direct” hypothesis.

Answer: We appreciate the reviewer’s valuable suggestion. As shown in Supplementary fig. 3c, the level of RBOHD protein was not significantly different from 0 to 30 min after ATP treatment. Our “direct” hypothesis is based on “an elevation of ROS production upon ATP addition”.

7. The text (lines 261-263) indicates that growth of surface-inoculated DC3000 was significantly greater on *dorn1-3* and *rboh*d relative to Col-0. However, the ANOVA for figure 5e do not support this statement for *rboh*d. Is that statement based on alternate statistical testing? If not, it should be corrected.

Answer: Thanks for this comment. In fig 5e, growth of surface-inoculated Pst DC3000 with 100, 250, 500 μ M ATP “was significantly greater on *dorn1-3* and *rboh*d relative to Col-0”. We conducted the one-sided ANOVA analysis again and using $P < 0.05$, the results were significantly different from the *rboh*d mutant.

8. Figure 5f should include the *rboh*d mutant as a control. Are the infection assays of figure 5f carried out by the same method as figure 5e? Are they carried out in the absence of supplemental ATP? Assuming the answer to both questions is yes, then it is important to explain why the S22A and S22AT24A derivatives support increased bacterial growth that is not observed with the *rboh*d mutant in figure 1e.

Answer: Thanks for this comment. In figure 5f, we added the *rboh*d mutant as a control and to increase the significance we conducted the experiment with 200 μ M ATP.

9. The rapid induction by DC3000 of extracellular ATP, DORN1-dependent induction of ROS, and

DORN1 phosphorylation by DC3000 (Fig. 6) are each rapid responses, indicating that one or all may be PAMP-induced. The authors should repeat the experiments of Figure 6 with inclusion of the *hrcC*-derivative of DC3000, *flg22* and *elf18*.

Answer: We appreciate the reviewer's valuable suggestion. Many stresses can induce ATP release (Kim et al., 2006, plant physiology), such as chitin or even touch. Treating with DC3000 bacteria is also a stress. To address the reviewer's concerns, we now include data using the *hrcC* mutant and *flg22* in figure 6.

Minor edits:

- The sentence from line 70-72 seems to belong in the preceding paragraph.

Answer: Thanks. We revised it. However, it was a new description.

- Line 121 should read "DORN1-mediated" rather than "DORN1-mediate".

Answer: Thanks. We revised this.

- Perhaps it is better call it DORN1 self-association rather than homodimer formation, as there is no evidence distinguishing between dimers or higher order complexes.

Answer: Thanks. We revised "homodimer" to "self-association".

- On line 266, the reference to supplementary figure 3b does not pertain to the sentence.

Answer: Thanks. We revised it to supplementary fig 5c (relative to supplementary fig 3c in the last manuscript).

- Could the authors speculate on why ATP promotes stomatal closure in the light, but opening in the dark?

Answer: We appreciate the reviewer's valuable suggestion. However, our purpose is to investigate DAMP immunity signaling, so we didn't focus on the possible role of ATP under dark conditions.

Reviewer #3 (Remarks to the Author):

Extracellular ATP (eATP) is among the characterized DAMPs that can elicit immune responses in plants, and the LecRK DORN1 was previously identified as the Arabidopsis eATP receptor. In this manuscript, Chen et al present evidence that DORN1 can directly phosphorylate the NADPH oxidase RBOHD, which is responsible for producing the ROS burst that is characteristic of PTI signaling in response to many PAMPs/DAMPs. These findings are of great significance, as they would provide the first evidence for the direct trans-phosphorylation of RBOHD by a pattern recognition receptor (PRR). While the manuscript is well-written and the evidence general supports their main findings, the model presented by the authors is not entirely comprehensive and may require revision. These comments along with others are outlined below. Such issues aside, this manuscript provides a valuable contribution to our understanding of PTI signaling.

The model presented in Figure 6 seems overly simplified, as it suggests that direct transphosphorylation of RBOHD by DORN1 is sufficient for the complete activation of RBOHD downstream of DAMP (eATP) perception, however, this is not supported by the authors own data (Fig 5c). Also, since it was previously shown that the cytoplasmic kinase BIK1 phosphorylates RBOHD downstream of the Atpep receptor (PEPR1) and PAMP receptors (i.e. FLS2, EFR) (Kadota et al., Mol Cell 2014; Li et al., Cell Host Microbe 2014), it is essential that the authors test whether BIK1 is required for eATP-induced ROS production (as well as eATP-induced stomatal closure), and similarly whether mutations in BIK1-regulated RBOHD phosphosites affect these responses.

Answer: We appreciate the reviewer's valuable suggestion. Our model presented in fig 6e is **only** for "the mechanism by which ATP elicits DORN1-mediated RBOHD phosphorylation to regulate stomatal immunity", but not "completely activate", and we did not claim that "DORN1 is sufficient for the complete activation of RBOHD downstream of DAMP (eATP) perception" in our manuscript. So our data (such as Fig 5c) do support this ATP signaling model.

BIK1 and CPKs can phosphorylate RBOHD to regulate ROS, so we mention in our discussion that "At this point it remains unclear as to how or whether these various regulatory mechanisms interact to either antagonize or synergize the ROS response". We have some preliminary data suggesting a complex relationship between DORN1-RBOHD-BIK1-CPK5 but this clearly will require a significant amount of further work that we hope will make the basis of a future paper. As with all scientific studies, while you gain some answers you very often reveal even more questions. Our study is no different but clearly provides exciting, novel data concerning the role of ATP in stomatal control and innate immunity.

Furthermore, given these previous results and the fact that Atpep is a DAMP, it is an overstatement to suggest that PAMP perception activates RBOHD via BIK1, while DAMPs do so directly via their respective PRRs, as the authors currently suggest in their discussion.

Answer: Thanks for this comment. We didn't mention this idea in our manuscript, and ATP and Atpep are different DAMP factors with different receptors and, hence, it would not be unexpected that responses associated with these DAMPs would show differences.

Also, the fact that some eATP-induced responses are still observed in *dorn1-3* directly challenges the fact that DORN1 is the eATP receptor (Choi et al., Science 2014). It is not correct to state that the "presence of...multiple regulatory pathways that control RBOHD actively likely explains why *dorn1-3* mutant plants still retain some, residual ROS production" (lines 347-349). If DORN1 is indeed the eATP receptor, then, all regulatory pathways and downstream signals triggered in response to eATP treatment should be abolished (if *dorn1-3* is a null mutant or a complete loss-of-function mutant). Based on these comments, it is also not correct to state in the legend of Figure 2 that DORN is "essential for the ATP-induced ROS burst" (lines 772-773) given the substantial ATP-induced ROS burst in *dorn1-3*.

Answer: Thanks for asking this valuable question. The reviewer is making extrapolations that we did not intend. Clearly *dorn1* mutants are significantly affected in their ROS response and, hence, we feel it is appropriate to conclude that DORN1 is 'essential' for this response, albeit that some activity remains. We do believe that some of this could be due to other, complex pathways that are at work which control ROS production, including RBOHD activity. However, it is also quite possible, indeed probable, that other, unidentified ATP receptors may be involved. There are several P2X and P2Y receptors in animals and we fully expect that this is likely the case in plants. A clear correlation can be seen with the human P2X7 receptor where a single mutant in this gene retains some ROS production (such as Figure 3, Pfeiffer et al., 2007). There is an active effort in our lab to identify additional ATP receptors. What is clear from our publications and the current paper is that, under the conditions of our assays, DORN1 is the primary ATP receptor. We agree that under different conditions (e.g., plants of different ages, developmental stages, tissue-specific responses, etc.) the activity of other ATP receptors may be of importance. Unfortunately, we currently lack any data to make this case beyond speculation.

See: "Pfeiffer, Z. A. et al. Nucleotide receptor signaling in murine macrophages is linked to reactive oxygen species generation. Free Radic. Biol. Med. 42, 1506-1516, (2007)".

Appropriate negative controls for BiFc assays are lacking, as it is not surprising that an unfused YFPc does not reconstitute a BiFc signal when expressed with a protein-YFPc fusion. See this recent commentary about appropriate controls for BiFc experiments: <http://www.plantcell.org/content/28/5/1002.long>.

Answer: Thanks for pointing out this issue. We added full-length CERK1 as a negative control in fig 3b.

Minor comments:

- Lines 92-98: the authors only provide a partial explanation to the observations made previously by Clark et al. (2011), as none of their data explains why eATP stimulates stomatal opening in darkness. Please rephrase accordingly.

Answer: We appreciate the reviewer's valuable suggestion. Our purpose is to investigate DAMP immunity signaling, so we didn't address the possible function of ATP under dark conditions.

- Fig 1e: the IP sample should be shown along with coIP to show relative enrichment

Answer: we already showed IP sample in the "Input" of fig 1e in the previous submission.

- Line 151: it is not clear with the authors only used the kinase domain of CERK1 here, while full-length DORN1 was otherwise used.

Answer: Thanks. Here, we used full-length CERK1.

- Fig 3b: the FM64 PM marker in the YFP control clearly co-localizes with cytosolic YFP signal, which suggests the signal is not specific.

Answer: Thanks. In the previous submission, the FM64 marker had clear nuclear but not YFP signal, so it was not co-localized with cytosolic YFP signal. Different intensity and focus of separated fluorescence will affect the merged figure. However, we changed it in the new manuscript just to make it clearer.

- Fig 4b: The evidence that RBOHD-N transphosphorylation is different on the three S/T A mutants is not convincing - was this experiment repeated?

Answer: Thanks. Yes, this experiment was actually repeated on three different occasions by three different scientists in the lab.

- Line 242: where are the data showing that the "ATP-dependent pathway is independent of ABA-induced stomatal closure? This hasn't been directly tested.

Answer: In fig 5a, b, ATP treatment cannot stimulate stomatal closure in *dorn1-3* and *rbohhd* mutants, while ABA induced stomatal closure. So "ATP-dependent pathway is independent of ABA-induced stomatal closure".

- Fig 6b: "Quantification of eATP concentration" is not accurate – the RLU value does allow for the calculation of concentration.

Answer: Thanks. We revised it to "Relative quantification of eATP concentration".

- Fig S3b: this suggests that no other PAMPs/DAMPs modulate stomatal aperture, as *dorn1-3* mutants are nearly completely insensitive to DC3000 cor- - what about flagellin, etc, which have been previously shown to induce stomatal closure during infection (Zeng & He, Plant Physiol 2010)?

Answer: Thanks. The *flg2* mutant shows complete insensitive to DC3000 COR-, but *efr-1* showed no difference from WT, indicating that different receptors have different functions in response to DC3000 COR-.

- Fig S3c: the different constructs are clearly expressed at different levels

Answer: Thanks, we re-quantized the loading proteins of each sample and this is now shown.

Reviewers' comments:

Reviewer #1 (Remarks to the Author):

In the revised manuscript "Extracellular ATP elicits DORN1-mediated RBOHD phosphorylation to regulate stomatal aperture" by Chen et al., the mass spectrometric-based approach is more clearly described for the reader and helps to facilitate acceptance of stated conclusions. The authors took the time to improve data availability and context for results through the inclusion of PRIDE data deposit and supplementary tables/data that are helpful in resolving our previous reservations for publication. Specifically, they improved the description in methods for peptide scoring and inclusion of phosphosite localization for each peptide (supplementary data).

However, the now included supplementary data could be more clearly presented. For example, there is no heading or description for the table and the layout is not intuitive. Additionally, we have the following comments:

For supplementary table one: helpful for phosphopeptide column to denote where phosphorylation is occurring. For example, instead of GFSKDEFLGK, do GFS(p)KDEFLGK. Minor note: phosphopeptide is spelled incorrectly in header of table.

For Introduction: define abbreviation "eATP"

In Results under "DORN1 autophosphorylation is essential for downstream signaling": last sentence "self-association" should be "self-associate"

In Results under "RBOHD is a kinase substrate of DORN1", last sentence of first paragraph: Make more clear that phosphorylation was seen on a synthetic peptide in the KiC assay, which corresponds to a peptide in the protein RBOHD.

In Results under "DORN1 phosphorylates RBOHD at Ser22 and Thr24 to promote ROS production", line 6: "DORN1-KD-2 failed to phosphorylated" should be "DORN1-KD-2 failed to phosphorylate"

In Methods under "Mapping of the autophosphorylation sites of DORN1": specify scoring parameters for peptide scoring and phosphosite localization.

Reviewer #2 (Remarks to the Author):

The authors have satisfactorily addressed my concerns.

Reviewer #3 (Remarks to the Author):

It is somehow disappointing that the authors were dismissive – both in their tone of their rebuttal letter but also in their unwillingness to perform some of the straightforward experiments – of the constructive suggestions made by this reviewer.

This reviewer, for example, does not challenge the finding that DORN1 directly phosphorylates RBOHD on S22 and that this regulation is important for ROS production in response to eATP treatment. However, given the important role of BIK1 downstream of multiple PRRs involved in the perception of several PAMPs (e.g. flg22, elf18, chitin) and DAMPs (e.g. Pep1), this reviewer still feels that it is important that the authors test whether eATP-induced ROS production also involves BIK1. This should

be easily feasible given the availability of published *bik1* (and *bik1/pbl1*) mutants and transgenic lines expressing RBOHD mutants in key BIK1-mediated phosphosites. The inclusion of such data are important as, if negative, they would indeed indicate that eATP-induced ROS production solely relies on DORN1-mediated direct phosphorylation of RBOHD; while, if negative, they would indicate that eATP-induced ROS production involves both DORN1- and BIK1-mediated direct phosphorylation of RBOHD.

Also, despite what the authors wrote in their rebuttal, lines 356-357 on page 17 still suggest they claim that PAMPs directly regulate RBOHD via BIK1, while DAMPs would do so via DORN1. This statement is firstly not supported by the previous demonstration that Pep1-triggered ROS production involves BIK1-mediated RBOHD phosphorylation, but secondly is also not supported experimentally until the authors will have tested the potential role of BIK1 in eATP-triggered ROS production. It seems – based on their rebuttal – that the authors actually already have these data.

This reviewer also still take issue with the statement that DORN1 is “essential” for eATP perception and response. “Essential” means absolutely required, which the data indicates is clearly not the case here, given the significant (~50%) ROS burst elicited in the *dorn1-3* background. At the very least the text should be altered to reflect this, and could incorporate parts of the authors’ response to state clearly that additional plant eATP receptors exist. The model should also account for the hypothesis that other receptor(s) contribute significantly to the eATP-triggered ROS burst.

In a similar vein, did the authors detect any difference in the timing of eATP-responsive ROS in the *dorn1-3* vs WT backgrounds (e.g. curves vs total ROS burst values)?

As previously requested, the authors should show the IP blots for Figures 1e, 2b, 3a and 3c. “IP” and “Input” are not the same thing, and it is important to demonstrate that an equal amount of bait has been IP-ed for comparison; especially when to claims of ligand-dependent differences are made.

There is still no data showing that the eATP-dependent pathway is independent of ABA-induced stomatal closure. The authors answered: “In fig 5a, b, ATP treatment cannot stimulate stomatal closure in *dorn1-3* and *rbohD* mutants, while ABA induced stomatal closure. So “ATP-dependent pathway is independent of ABA-induced stomatal closure”. These results however only show that ABA-induced stomatal closure is independent of DORN1, which is different! Thus, if the authors would like to claim convincingly that eATP-induced stomatal closure is independent of ABA, they should perform eATP treatment of ABA biosynthetic/signaling mutants.

The authors’ response to the comment on the function of eATP under dark conditions is fine (that they are exclusively interested in DAMP responses), however, they should explain (speculate?) how their findings may fit into the contrasting light/dark effects of ATP application, as suggested by two reviewers – this is important since the authors themselves introduce this phenomenon in the Introduction... Furthermore, have they tested whether eATP stimulates the same ROS burst when leaf discs are kept in light vs dark conditions overnight?

In Figure 6d, the authors can only refer to “DORN1 protein phosphorylation” if they could show that the observed upper band in response to eATP, Pst DC3000 or Pst *hrcC* is affected by treatment with a general protein phosphatase. In the absence of such data, the authors can only refer to “DORN1 protein band shift” and speculate that this band shift may be due to protein phosphorylation.

Figure 4d would benefit from an Anova statistical test, and all graphs should present statistical analysis.

Page 10, line 215: why “remarkably”?

Reviewers' comments:

Reviewer #1 (Remarks to the Author):

In the revised manuscript “Extracellular ATP elicits DORN1-mediated RBOHD phosphorylation to regulate stomatal aperture” by Chen et al., the mass spectrometric-based approach is more clearly described for the reader and helps to facilitate acceptance of stated conclusions. The authors took the time to improve data availability and context for results through the inclusion of PRIDE data deposit and supplementary tables/data that are helpful in resolving our previous reservations for publication. Specifically, they improved the description in methods for peptide scoring and inclusion of phosphosite localization for each peptide (supplementary data).

However, the now included supplementary data could be more clearly presented. For example, there is no heading or description for the table and the layout is not intuitive. Additionally, we have the following comments:

For supplementary table one: helpful for phosphopeptide column to denote where phosphorylation is occurring. For example, instead of GFSKDEFLGK, do GFS(p)KDEFLGK. Minor note: phosphopeptide is spelled incorrectly in header of table.

Answer: As suggested, we added (p) to all phosphopeptide in supplementary table one.

For Introduction: define abbreviation “eATP”

Answer: We define “eATP” as “ATP can also be released into the extracellular matrix, where it is referred to as extracellular ATP (eATP)” in the first paragraph of Introduction.

In Results under “DORN1 autophosphorylation is essential for downstream signaling”: last sentence “self-association” should be “self-associate”

Answer: We revised it, thank you.

In Results under “RBOHD is a kinase substrate of DORN1”, last sentence of first paragraph: Make more clear that phosphorylation was seen on a synthetic peptide in the KiC assay, which corresponds to a peptide in the protein RBOHD.

Answer: We describe it now as “One of the most interesting of these proteins was RBOHD where the synthetic peptide GILRGANS(p)DT(p)NSDTEI was phosphorylated by DORN1-KD in the KiC assay (Fig. 2a)”.

In Results under “DORN1 phosphorylates RBOHD at Ser22 and Thr24 to promote ROS production”, line 6: “DORN1-KD-2 failed to phosphorylated” should be “DORN1-KD-2 failed to phosphorylate”

Answer: We revised it, thank you.

In Methods under “Mapping of the autophosphorylation sites of DORN1”: specify scoring parameters for peptide scoring and phosphosite localization.

Answer: As suggested, we describe them more specifically in the Methods under “Mapping of the autophosphorylation sites of DORN1”.

Reviewer #2 (Remarks to the Author):

The authors have satisfactorily addressed my concerns.

Answer: Thank you very much.

Reviewer #3 (Remarks to the Author):

It is somehow disappointing that the authors were dismissive – both in their tone of their rebuttal letter but also in their unwillingness to perform some of the straightforward experiments – of the constructive suggestions made by this reviewer.

Answer: We assure the reviewer that in no way did we intend to be dismissive of any of the criticisms of the paper from this reviewer or others. What we are presenting is a very data rich, focused paper that explains how DORN1 directly activates RBOHD at a biochemical level. No paper, including ours, can answer every question and, indeed, the best papers often raise more questions than they answer. We respect the reviewer's opinion and have thought of some of the same questions he/she raised in their critique. However, we believe that these are questions for the future and are not appropriate for the current paper, either because they are ancillary to the story we are telling and/or because the results of the experiments would not materially impact what we are presenting. Rest assured that our interests in DORN1 function are continuing and many, if not all, of the questions that the reviewer raises will be answered in time and published as separate papers when we feel that they are complete.

This reviewer, for example, does not challenge the finding that DORN1 directly phosphorylates RBOHD on S22 and that this regulation is important for ROS production in response to eATP treatment. However, given the important role of BIK1 downstream of multiple PRRs involved in the perception of several PAMPs (e.g. flg22, elf18, chitin) and DAMPs (e.g. Pep1), this reviewer still feels that it is important that the authors test whether eATP-induced ROS production also involves BIK1. This should be easily feasible given the availability of published *bik1* (and *bik1/pbl1*) mutants and transgenic lines expressing RBOHD mutants in key BIK1-mediated phosphosites. The inclusion of such data are important as, if negative, they would indeed indicate that eATP-induced ROS production solely relies on DORN1-mediated direct phosphorylation of RBOHD; while, if positive, they would indicate that eATP-induced ROS production involves both DORN1- and BIK1-mediated direct phosphorylation of RBOHD. Also, despite what the authors wrote in their rebuttal, lines 356-357 on page 17 still suggest they claim that PAMPs directly regulate RBOHD via BIK1, while DAMPs would do so via DORN1. This statement is firstly not supported by the previous demonstration that Pep1-triggered ROS production involves BIK1-mediated RBOHD phosphorylation, but secondly is also not supported experimentally until the authors will have tested the potential role of BIK1 in eATP-triggered ROS production. It seems – based on their rebuttal – that the authors actually already have these data.

Answer: The reviewer is correct that BIK1 could be playing a role in the action of DORN1. However, so could a variety of other proteins, as well as over all calcium levels, perhaps unknown

calcium dependent protein kinases, etc. The published literature clearly indicates that the regulation of RBOHD is complex reflecting the critical role that ROS production plays in a variety of processes. DORN1, regardless of any role for BIK1, clearly interacts directly with RBOHD, phosphorylates it and regulates its activity. That is the point of this paper. Jumping off into studies of BIK1 represents in our mind a separate study and, while clearly of possible importance, would not materially change either the findings or conclusions of this paper. Moreover, depending on the results obtained, it should be clear to the reviewer that what they are suggesting is not really 'easily feasible' since any results obtained would need significant validation, as well as follow on experiments to provide the proper context and to place the findings within the overall cellular signaling network (i.e., what they are really suggesting is an in depth, lengthy and costly, in manpower, study). Clearly, cellular regulation pathways are complex and our knowledge of this complexity is the result of step-by-steps studies that incrementally reveal this complexity and place it within the context of overall cell function. In this vein, we feel that our paper is a significant contribution and, hence, should stand on its own.

Because Pep1-triggered ROS production involves BIK1-mediated RBOHD phosphorylation, we revised the text to "This adds a new element to the overall regulation of RBOHD allowing for integration of responses to PAMPs (via BIK1), DAMPs (via DORN1 and BIK1) and a variety of stresses (via calcium signaling)".

This reviewer also still take issue with the statement that DORN1 is "essential" for eATP perception and response. "Essential" means absolutely required, which the data indicates is clearly not the case here, given the significant (~50%) ROS burst elicited in the *dorn1-3* background. At the very least the text should be altered to reflect this, and could incorporate parts of the authors' response to state clearly that additional plant eATP receptors exist. The model should also account for the hypothesis that other receptor(s) contribute significantly to the eATP-triggered ROS burst.

In a similar vein, did the authors detect any difference in the timing of eATP-responsive ROS in the *dorn1-3* vs WT backgrounds (e.g. curves vs total ROS burst values)?

Answer: The dictionary defines "essential" as "absolutely necessary, extremely important". Our use of the word refers more to the point of "extremely important", although we understand that the reviewer is looking at this with regard to being "absolutely necessary". However, it would do us little good to argue semantics when we are trying to convey that DORN1 is very important for the plant with regard to its respond to eATP.

Hence, we have modified the text to indicate that the data clearly show that DORN1 is "important" for the plant response. Although we are convinced that other eATP receptors exist in *Arabidopsis*, at the moment, we can provide no publishable data to show this. Hence, we are hesitant to speculate about this in the text. Moreover, as indicated above, it is clear from the literature that RBOHD regulation is complex and, hence, it may very well be that the remaining ROS burst in the *dorn1* mutant is being mediated via some other, unknown regulatory pathway.

As shown in the supplemental figure 3a, the curves of eATP-induced ROS in the *dorn1-3* mutant and WT are similar as the total ROS burst values (Fig. 2c).

As previously requested, the authors should show the IP blots for Figures 1e, 2b, 3a and 3c. “IP” and “Input” are not the same thing, and it is important to demonstrate that an equal amount of bait has been IP-ed for comparison; especially when to claims of ligand-dependent differences are made.

Answer: Thanks for pointing out this issue. We now show the IP blots in these figures.

There is still no data showing that the eATP-dependent pathway is independent of ABA-induced stomatal closure. The authors answered: “In fig 5a, b, ATP treatment cannot stimulate stomatal closure in *dorn1-3* and *rbohD* mutants, while ABA induced stomatal closure. So “ATP-dependent pathway is independent of ABA-induced stomatal closure”. These results however only show that ABA-induced stomatal closure is independent of DORN1, which is different! Thus, if the authors would like to claim convincingly that eATP-induced stomatal closure is independent of ABA, they should perform eATP treatment of ABA biosynthetic/signaling mutants.

Answer: We appreciate the reviewer’s valuable suggestion. As suggested, we checked the mutants of ABA receptors (PYRABACTIN RESISTANCE1 [PYR1], PYRABACTIN RESISTANCE-LIKE1 [PYL1], PYL2, and PYL4) and the downstream of ABA perception kinase protein OST1 (Open Stomata1), which are insensitive to ABA-induced stomatal closure. Both *ost1* and *pyr1/pyl1/pyl2/pyl4* quadruple were able to close stoma as wild type in response to ATP treatment (Supplementary Fig. 5b), indicating that the eATP-dependent pathway is independent of ABA-induced stomatal closure. This was a very reasonable suggestion by the reviewer that we felt was appropriate and materially improved the manuscript and, hence, we were happy to conduct this experiment. We thank the reviewer for the suggestion.

The authors’ response to the comment on the function of eATP under dark conditions is fine (that they are exclusively interested in DAMP responses), however, they should explain (speculate?) how their findings may fit into the contrasting light/dark effects of ATP application, as suggested by two reviewers – this is important since the authors themselves introduce this phenomenon in the Introduction. Furthermore, have they tested whether eATP stimulates the same ROS burst when leaf discs are kept in light vs dark conditions overnight?

Answer: We appreciate the reviewer’s suggestion. The ROS bursts triggered by the elicitors we tested (flg22, chitin, and ATPγs) in the wild type plant leaf discs overnight in darkness are significantly higher than in the light (see figure included). This suggests that the effect of light may be at the level of ROS production and not necessarily something related to the specific elicitor. However, beyond this experiment, as we stated, we really have not addressed the issues of light effects and, hence, are very reluctant to speculate on something that could turn out to be very complex.

In Figure 6d, the authors can only refer to “DORN1 protein phosphorylation” if they could show that the observed upper band in response to eATP, Pst DC3000 or Pst *hrcC* is affected by treatment with a general protein phosphatase. In the absence of such data, the authors can only refer to “DORN1 protein band shift” and speculate that this band shift may be due to protein

phosphorylation.

Answer: Thanks for pointing out this issue. We refer to “DORN1 protein modification, likely due to protein phosphorylation.” both in the figure and text.

Figure 4d would benefit from an Anova statistical test, and all graphs should present statistical analysis.

Answer: We appreciate the reviewer's valuable suggestion. We re-analyzed the data using an Anova statistical test, see the figures and legends.

Page 10, line 215: why “remarkably”?

Answer: We removed the offending word, thank you.

REVIEWERS' COMMENTS:

Reviewer #1 (Remarks to the Author):

In the revised manuscript "Extracellular ATP elicits DORN1-mediated RBOHD phosphorylation to regulate stomatal aperture" by Chen et al., the authors took the time to address most of the concerns regarding data availability and ease of understanding supplementary data. While this is appreciated, I believe the supplementary data could still be explained more prior to publication, as mentioned in the previous comments for the authors. For instance, there is only one mention of "supplementary data 1" in the text however there are 6 tabs of data. None of them have headings or explanation of acronyms used (PSM, ΔCn etc...). Additionally, there are numbers in columns C-K before the header in each tab that are never explained.

Reviewer #3 (Remarks to the Author):

I would like to thank the authors for their detailed response to my previous comments. I want to reassure the authors that I did not question the relevance of the report that DORN1 directly phosphorylates RBOHD and that this is important for ROS production in response to eATP. I also did not suggest that the authors should perform comprehensive studies on the potential additional role of BIK1 in this regulation, but I still think that it would have been nice to test simply whether *bik1* mutants are affected in eATP-induced ROS production, given the reported direct interaction of BIK1 with multiple PRRs and the reported role of BIK1 in RBOHD regulation (which is differently to testing the potential role of calcium or calcium-dependent protein kinases, as these regulatory components are not directly linked to PRRs). This experiment would have been very straight-forward (and much less time-consuming than the stomatal aperture assays performed in ABA signaling mutants). I now leave the decision to the editor to decide whether or not the authors should include such data (which they most likely already have).

Minor comment: it is incorrect to call the pull-down an "IP" in Fig 3a, as there is no antibody involved.

REVIEWERS' COMMENTS:

Reviewer #1 (Remarks to the Author):

In the revised manuscript “Extracellular ATP elicits DORN1-mediated RBOHD phosphorylation to regulate stomatal aperture” by Chen et al., the authors took the time to address most of the concerns regarding data availability and ease of understanding supplementary data. While this is appreciated, I believe the supplementary data could still be explained more prior to publication, as mentioned in the previous comments for the authors. For instance, there is only one mention of “supplementary data 1” in the text however there are 6 tabs of data. None of them have headings or explanation of acronyms used (PSM, ΔCn etc...). Additionally, there are numbers in columns C-K before the header in each tab that are never explained.

Answer: As suggested, we add the header and Foot notes for the explanation of acronyms in the “supplementary data 1”.

Reviewer #3 (Remarks to the Author):

I would like to thank the authors for their detailed response to my previous comments. I want to reassure the authors that I did not question the relevance of the report that DORN1 directly phosphorylates RBOHD and that this is important for ROS production in response to eATP. I also did not suggest that the authors should perform comprehensive studies on the potential additional role of BIK1 in this regulation, but I still think that it would have been nice to test simply whether *bik1* mutants are affected in eATP-induced ROS production, given the reported direct interaction of BIK1 with multiple PRRs and the reported role of BIK1 in RBOHD regulation (which is differently to testing the potential role of calcium or calcium-dependent protein kinases, as these regulatory components are not directly linked to PRRs). This experiment would have been very straight-forward (and much less time-consuming than the stomatal aperture assays performed in ABA signaling mutants). I now leave the decision to the editor to decide whether or not the authors should include such data (which they most likely already have).

Answer: Thank you for your suggestion.

Minor comment: it is incorrect to call the pull-down an “IP” in Fig 3a, as there is no antibody involved.

Answer: As suggested, we advise “IP” to “Pull-down”.